# AudioMarathon: A Comprehensive Benchmark for Long-Context Audio Understanding and Efficiency in Audio LLMs

## Abstract

Processing long-form audio is a major challenge for Large Audio Language models (LALMs). These models struggle with the quadratic cost of attention ($\mathcal{O}(N^2)$) and with modeling long-range temporal dependencies. Existing audio benchmarks are built mostly from short clips and do not evaluate models in realistic long context settings. To address this gap, we introduce **AudioMarathon**, a benchmark designed to evaluate both *understanding* and *inference efficiency* on long-form audio. AudioMarathon provides a diverse set of tasks built upon three pillars: long-context audio inputs with durations ranging from 90.0 to 300.0 seconds, which correspond to encoded sequences of 2,250 to 7,500 audio tokens, respectively, full domain coverage across speech, sound, and music, and complex reasoning that requires multi-hop inference. We evaluate state-of-the-art LALMs and observe clear performance drops as audio length grows. We also study acceleration techniques and analyze the trade-offs of token pruning and KV cache eviction. The results show large gaps across current LALMs and highlight the need for better temporal reasoning and memory-efficient architectures. We believe AudioMarathon will drive the audio and multimodal research community to develop more advanced audio understanding models capable of solving complex audio tasks.

## 1 Introduction

Multimodal Large Language Models (MLLMs) have demonstrated remarkable capabilities in understanding and processing various data modalities (Alayrac et al., 2022; Li et al., 2023; Liu et al., 2023; Chen et al., 2024b; Kang et al., 2025; Zhang et al., 2024a; Wen et al., 2024). With audio being a key area of advancement, the ability to comprehend spoken language, environmental sounds, and music has opened up new frontiers for applications ranging from advanced speech recognition (Radford et al., 2023) to sophisticated audio-based reasoning (Borsos et al., 2023).

However, a significant and persistent challenge remains: the effective processing of long-form audio inputs. As the duration of audio increases, Large Audio Language Models (LALMs) face a dual challenge of escalating computational and memory costs (Vaswani et al., 2017), coupled with the inherent difficulty of capturing and modeling extended temporal dependencies (Beltagy et al., 2020; Zaheer et al., 2020). This bottleneck severely limits their practical application in real-world scenarios such as analyzing meetings, podcasts, or extended dialogues. A major factor hindering progress in this domain is the lack of comprehensive benchmarks designed to evaluate the long audio capabilities of LALMs rigorously. Existing audio benchmarks predominantly consist of short clips, typically only a few seconds long (Weck et al., 2024; Sakshi et al., 2024; Yang et al., 2024; Wang et al., 2024a). While valuable, these benchmarks fail to assess a model's ability to maintain coherence, reason over long time spans, and manage computational resources efficiently when faced with minute-scale and even hour-scale audio inputs. This gap leaves a critical aspect of model performance unevaluated and obstructs the development of more robust and scalable audio understanding systems.

To address this critical gap, we introduce **AudioMarathon**, a comprehensive audio benchmark meticulously designed to evaluate LALMs on long-context audio understanding and inference efficiency. AudioMarathon is built on three foundational pillars: ❶ **Long-form Audio Context**, featuring audio durations ranging from 90.0 to 300.0 seconds to simulate realistic scenarios; ❷ **Full**

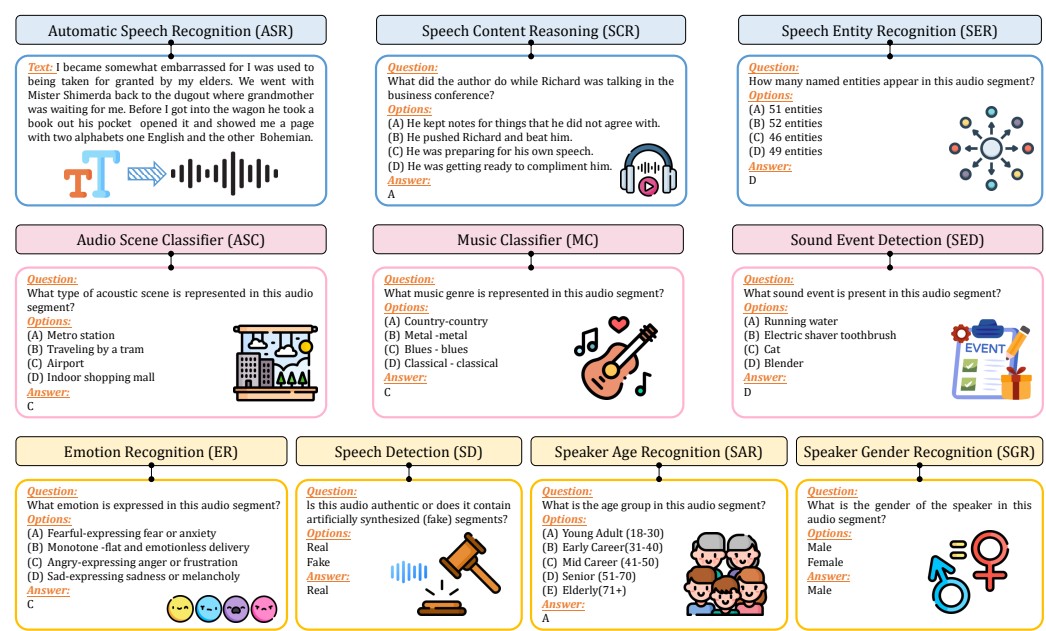

Figure 1: Overview of the AUDIOMARATHON. AUDIOMARATHON extends short audio clips to long-form audio with a diverse range of task categories, offering a comprehensive and practical assessment of audio intelligence in real-world scenarios.

**Domain Coverage**, encompassing a diverse range of audio types including speech, environmental sounds, and music, as well as comprehensive task coverage spanning ten representative sub-tasks (ASR, SCR, SER, MC, ASC, SED, ER, SD, SAR, SGR) across Speech Context Understanding, Audio Scene Understanding, and Voice Characteristic Identification; and ❸ **Complex Reasoning**, incorporating multi-hop inference tasks that require models to connect disparate pieces of information across extended temporal windows.

Beyond just establishing a challenging new benchmark, this work also investigates crucial aspects of inference efficiency for long audio. We systematically evaluate a suite of state-of-the-art Audio LLMs (Chu et al., 2023; Abouelenin et al., 2025; Xu et al., 2025a), analyzing their performance degradation as input length increases. Furthermore, we explore and quantify the effectiveness and trade-offs of various cost-reduction strategies, including inference-time **Token pruning** (Chen et al., 2024a; Zhang et al., 2024b; Wen et al., 2025b) and **KV-cache eviction** techniques (Li et al., 2024b). Our findings reveal substantial performance gaps among current models in long-context scenarios and underscore the pressing need for improved temporal reasoning and memory-efficient processing.

By providing a unified and challenging evaluation suite, we aim to catalyze future research. We release AUDIOMARATHON to the community to foster the development of the next generation of scalable, efficient, and robust LALMs capable of truly understanding the rich, continuous tapestry of the auditory world. Our main contributions are summarized as follows:

- AUDIOMARATHON is presented as a comprehensive benchmark for long audio understanding, characterized by extended audio durations, diverse domain coverage, and complex reasoning tasks.
- Our work thoroughly evaluates state-of-the-art LALMs on AUDIOMARATHON, revealing the specific challenges encountered when processing long audio inputs.
- In addition, we systematically analyze various inference efficiency techniques, such as token pruning and KV-cache eviction, to quantify their effectiveness and trade-offs.

## 2 AUDIOMARATHON

### 2.1 OVERVIEW

Existing audio benchmarks predominantly comprise short audio clips, often only a few seconds, thereby failing to capture the complexity of real-world scenarios such as meetings, podcasts, and

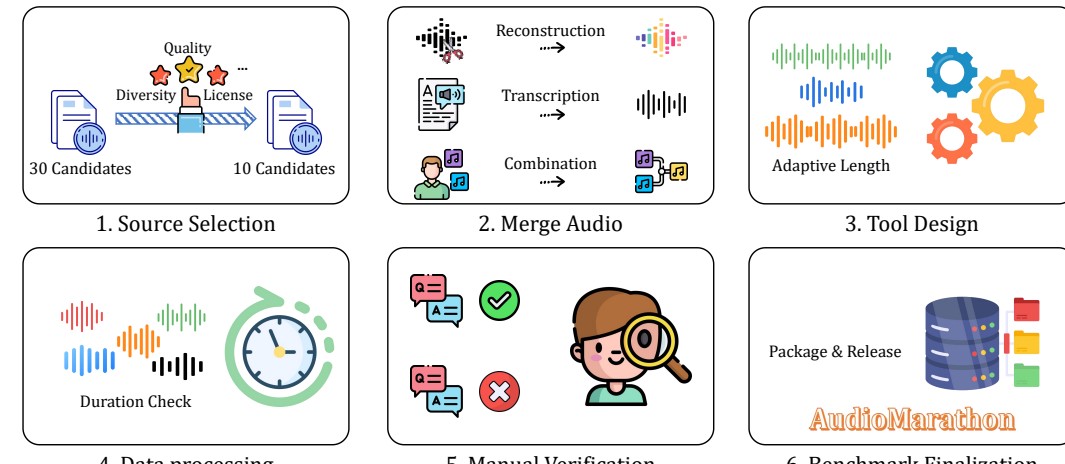

Figure 2: The six-stage data pipeline for constructing the AUDIOMARATHON

extended dialogues. To close the pronounced gap in benchmarks for long-form audio understanding, we present **AUDIOMARATHON**, a comprehensive suite designed to evaluate the advanced capabilities of LALMs. The construction of AUDIOMARATHON follows a rigorous six-stage pipeline (Figure 2), ensuring diversity, difficulty, and high annotation quality. Figure 4 and Table 5 summarize the final composition across task categories, while Table 1 and Table 2 compare AUDIOMARATHON against existing benchmarks.

## 2.2 DATA COLLECTION AND ANNOTATION

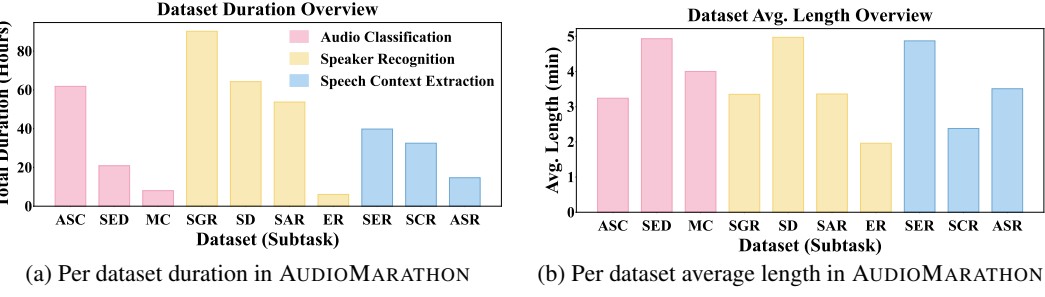

(a) Per dataset duration in AUDIOMARATHON  (b) Per dataset average length in AUDIOMARATHON

Figure 3: Per dataset duration and average length in AUDIOMARATHON

We adopt a rigorous multi-stage framework to construct **AUDIOMARATHON**, detailed below.

**Step 1. Source Selection.** From 30 candidate datasets, we selected ten subsets according to task coverage and acoustic diversity. The tasks are grouped into three categories: Speech Context Understanding, Audio Scene Understanding, and Voice Characteristic Identification, ensuring both practical relevance and expert-level reasoning challenges.

**Step 2. Merge Audio.** Considering the characteristics of different tasks, we designed specific concatenation logic to merge individual clips into longer sequences.

**Step 3. Tool Design.** We developed a custom concatenation script to automate merging process. The tool flexibly supports generating sequences of variable length within the constraints of the source material, allowing to adapt sequence duration for different experimental settings.

**Step 4. Data Processing.** Each audio file was paired with a task-specific prompt and multiple-choice options. Option generation followed customized strategies tailored to each task, and the implementation has been released as open source. Model predictions

Figure 4: Task composition of AU-DIOMARATHON by category

Table 1: Comparison of audio datasets in terms of duration, size, average audio length, and domain coverage (speech, sound, and music).

| Tasks | Duration | Size | Domain | | | Average audio duration |
| --- | --- | --- | --- | --- | --- | --- |
| | | | Speech | Sound | Music | |
| MuChoMusic (Weck et al., 2024) | 5.1h | 1.1k | ✗ | ✗ | ✓ | 25.7 sec |
| BLAB (Ahia et al., 2025) | 833h | 1.6k | ✓ | ✗ | ✗ | 51.0 min |
| MMAR (Ma et al., 2025) | 5.5h | 1k | ✓ | ✓ | ✓ | 19.4 sec |
| MMSU (Wang et al., 2025) | 9.73h | 5k | ✓ | ✗ | ✗ | 7.0 sec |
| MMAU (Sakshi et al., 2024) | 28.16h | 10k | ✓ | ✓ | ✓ | 10.1 sec |
| AIR-Bench (Yang et al., 2024) | 251.6h | 21k | ✓ | ✓ | ✓ | 35.2 sec |
| AudioBench (Wang et al., 2024a) | 400h | 100k | ✓ | ✓ | ✓ | 14.0 sec |
| **AudioMarathon** *(ours)* | 392h | 6.6k | ✓ | ✓ | ✓ | 212.8 sec |

were evaluated by two criteria: (i) exact match to a provided choice, or (ii) inclusion of the complete correct option without any extraneous information.

**Step 5. Manual Verification.** To ensure data quality, 10% (at least 20) samples per sub-dataset were randomly reviewed using the criteria detailed in Appendix D. Any dataset failing inspection was reconstructed and revalidated until all checked samples passed.

**Step 6. Benchmark Finalization.** From the fully annotated QA pairs, 6,567 instances were selected to ensure balanced coverage of all 10 tasks and audio types. The concatenated files had durations from 90.0 to 300.0 seconds, balancing long-context evaluation with computational feasibility.

## 2.3 COMPARISON WITH OTHER BENCHMARKS

Table 2: Comparison of AUDIOMARATHON with existing audio understanding and reasoning benchmarks across key properties and capabilities.

| Capability | AUDIOMARATHON | MuChoMusic | BLAB | MMAR | MMSU | MMAU | AIR-Bench | AudioBench |
| --- | --- | --- | --- | --- | --- | --- | --- | --- |
| Long Audio Understanding | ✓ | ✗ | ✓ | ✗ | ✗ | ✗ | ✗ | ✗ |
| Full Domain Coverage | ✓ | ✗ | ✗ | ✓ | ✓ | ✓ | ✓ | ✗ |
| Multi-Hop Inference | ✓ | ✗ | ✗ | ✗ | ✗ | ✗ | ✗ | ✗ |
| Speaker attribute coverage | ✓ | ✗ | ✗ | ✗ | ✓ | ✗ | ✓ | ✓ |
| Contain deepfake audio | ✓ | ✗ | ✗ | ✗ | ✗ | ✗ | ✓ | ✗ |
| Complex task hierarchy | ✓ | ✗ | ✗ | ✓ | ✓ | ✓ | ✓ | ✗ |
| Emotional and Semantic Understanding | ✓ | ✗ | ✓ | ✓ | ✓ | ✓ | ✓ | ✓ |

**Long Audio Understanding.** Public audio benchmarks mostly use second-level clips (e.g., MMAR 19.4 s, MMAU 10.1 s, MMSU 7.01 s, AudioBench 14 s), which miss minute-scale complexity. BLAB includes long audio (∼51.0 min) but is speech-centric. AUDIOMARATHON targets realistic long-form use with durations ranging from 90.0 to 300.0 seconds and supports flexible duration control.

**Full Domain Coverage.** Audio spans three domains: speech, sound, and music. Most benchmarks cover one or two, limiting cross-domain robustness. Our proposed AUDIOMARATHON covers all three with balanced sampling for comprehensive evaluation and cross-domain studies.

**Multi-Hop Inference.** We include an audio version of RACE generated via Text-to-Speech (Kokoro-82M (Nayak, 2025)), preserving RACE's multi-hop reasoning while adding long-term acoustic dependencies—a stricter test of comprehension, memory, and reasoning.

## 3 EXPERIMENTS AND EVALUATIONS

**Models.** We compare 16 recent Large Audio Language Models (LALMs), including ten open-source models and six closed-source models. The open-source models are Phi-4-Multimodal (Abdin et al., 2024), Qwen2.5-Omni-3B (Xu et al., 2025a), and Aero-1-Audio (Li et al., 2025a). Phi-4-Multimodal and Qwen2.5-Omni-3B are multi-modal large language models, while Aero-1-Audio is a compact audio language model designed for audio-centered tasks. The proprietary models are from the Gemini family: Gemini-2.5-Pro (Comanici et al., 2025a), Gemini-2.5-Flash (Comanici et al., 2025a), Gemini-2.0-Flash (Comanici et al., 2025b), and GPT-4o. All are multi-modal models, with Gemini-2.5-Flash and Gemini-2.0-Flash optimized for faster inference.

**Evaluation Metrics.** Our evaluation considers two dimensions: task performance and inference efficiency. For task performance, we adopt standard metrics per task: F1-score for classification and Multiple-Choice Questions (MCQs), Word Accuracy Rate (WAR) for ASR, and macro F1-score for audio event detection to balance precision and recall across classes. Inference efficiency is assessed via latency and peak GPU memory usage. We also report speedup over a vanilla model.

Table 3: Performance comparison of models on AudioMarathon across tasks, grouped into Speech Content Extraction (SER, SCR, ASR), Audio Classification (SED, MC, ASC), and Speaker Information Modeling (SD, ER, SAR, SGR). The Avg. column shows the mean score across all tasks. Best scores are in **bold**, second-best are underlined.

| Models | Speech Content Extraction | | | Audio Classification | | | Speaker Information Modeling | | | | Avg. |
|---|---|---|---|---|---|---|---|---|---|---|---|
| | SER | SCR | ASR | SED | MC | ASC | SD | ER | SAR | SGR | |
| **Open-source Audio LLMs** | | | | | | | | | | | |
| Phi-4-Multimodal | 18.4 | 69.3 | 92.7 | 55.1 | 46.7 | 23.4 | 26.4 | 27.3 | 26.6 | 91.1 | 47.7 |
| Qwen2.5-Omni-3B | 25.2 | 82.3 | 94.7 | 70.2 | 97.4 | 69.3 | 67.3 | 39.6 | 29.1 | 97.2 | 67.2 |
| Qwen2.5-Omni-7B | 26.3 | **85.1** | **98.1** | **78.4** | **100.0** | **72.2** | **72.3** | 53.4 | 21.4 | 98.0 | **70.5** |
| Audio-Flamingo-2 | 26.8 | 39.8 | 1.0 | 27.1 | 66.8 | 29.7 | 45.9 | 13.1 | 20.3 | 85.1 | 35.6 |
| Audio-Flamingo-3 | 21.7 | 78.9 | 94.3 | 59.5 | 97.0 | 54.1 | 33.7 | **54.3** | **40.7** | 96.2 | 63.0 |
| Gemma-3n-E2B-it | 22.5 | 51.6 | 91.3 | 50.2 | 56.8 | 28.2 | 35.1 | 15.2 | 12.2 | 91.6 | 45.5 |
| Gemma-3n-E4B-it | 19.0 | 56.9 | 93.2 | 50.2 | 71.9 | 31.7 | 35.9 | 18.9 | 21.8 | 93 | 49.3 |
| Voxtral-Mini-3B-2507 | 24.3 | 71.1 | 96.8 | 71.0 | 83.8 | 27.2 | 68.0 | 29.7 | 30.7 | 71.0 | 57.4 |
| Baichuan-Omni-1.5 | 12.4 | 11.2 | 86.5 | 45.7 | 52.0 | 25.8 | 49.2 | 18.9 | 10.2 | 81.5 | 39.3 |
| Aero-1-Audio | 17.9 | 56.6 | 43.7 | 55.0 | 83.9 | 39.9 | 33.7 | 32.0 | 17.8 | 47.5 | 42.8 |
| **Closed-source Audio LLMs** | | | | | | | | | | | |
| GPT-4o-Audio (Preview 2024-10-01) | 25.8 | 61.4 | 94.4 | 50.7 | 59.5 | 40.8 | 32.5 | 22.5 | 17.2 | 69.2 | 47.4 |
| GPT-4o-Audio (Preview 2024-12-17) | 25.7 | 60.2 | 94.7 | 51.2 | 67.6 | 41.9 | 30.8 | 21.8 | 19.9 | 73.1 | 48.7 |
| Gemini-2.0-Flash-Lite | 23.7 | 65.6 | 97.4 | 60.9 | 86.9 | 43.4 | 34.5 | 17.3 | 19.0 | 82.1 | 53.1 |
| Gemini-2.0-Flash | **30.9** | 71.8 | 96.4 | 68.1 | 88.5 | 54.1 | 32.1 | 20.1 | 39.2 | 93.1 | 59.4 |
| Gemini-2.5-Flash-Lite | 30.3 | 64.0 | 96.5 | 68.0 | 64.8 | 36.8 | 33.9 | 14.6 | 19.6 | 77.9 | 50.6 |
| Gemini-2.5-Flash | 28.1 | 83.6 | 96.8 | 69.2 | 79.3 | 40.8 | 33.1 | 31.9 | 34.3 | **99.3** | 59.6 |
| Human Evaluation | 45.1 | 88.1 | – | 96.2 | 100.0 | 100.0 | 100.0 | 90.8 | 71.4 | 97.0 | 87.6 |

**Evaluation Setup.** To conduct the ASR task, evaluations are performed on *test* subset of LibriSpeech-long (Park et al., 2024) after filtering. Except for ASR, all tasks are framed as MCQs with a single correct answer. SD and SGR provide two options, SAR provides five, and all other tasks use four. For each instance, the model receives the full audio along with an instruction-following prompt presenting a question and four labeled options. The model must select one option, and to mitigate positional bias, the option order is randomized.

## 4 EFFICIENCY OPTIMIZATION FOR LALMS

Processing extended audio sequences poses significant computational challenges for LALMs. A single 5-minute audio input can generate thousands of tokens, leading to quadratic memory growth and prohibitive inference latency (as shown in Table 11 of Appendix C). To address these bottlenecks, we systematically evaluate two complementary efficiency optimization strategies: **Token pruning** (Liu et al., 2025) during the prefilling stage and **KV-cache eviction**[1] during the decoding stage.

### 4.1 TOKEN PRUNING

Processing long-form audio sequences poses substantial memory and latency challenges for LALMs. One-minute audio input is embedded into 1500 tokens, requiring massive KV-cache storage and significantly slow decoding, thus making deployment impractical without compression. To address these bottlenecks, numerous approaches have emerged that directly reduce the number of tokens to improve inference efficiency. We evaluate four token pruning methods and four KV cache eviction strategies on our long-audio benchmark. Experiments are conducted on three open-source LALMs, including Qwen2.5-Omni-3B, Aero-1-Audio, and Phi-4-Multimodal.

We compare four token pruning strategies on AUDIOMARATHON. The baseline, Random pruning, discards tokens uniformly at random. FastV (Chen et al., 2024a) removes low-attention tokens, and DART (Wen et al., 2025b) applies redundancy-guided selection by discarding similar tokens. However, due to the strongly sequential nature of acoustic signals, naive or purely attention-based pruning can inadvertently remove brief phonetic cues or transient events, leading to degraded recognition. Unlike vision models, where redundancy often arises from spatial or semantic similarity, audio token redundancy primarily manifests as smooth temporal continuity. Therefore, we additionally design **Frame** as a time-aligned token pruning strategy to preserve rare or short-lived acoustic events that other methods may discard, making it a scheme tailored to audio characteristics.

---

[1]https://github.com/NVIDIA/kvpress

Table 4: Performance comparison of three open-source LALMs across token pruning methods and ratios on AudioMarathon tasks, grouped into Speech Content Extraction (SER, SCR, ASR), Audio Classification (SED, MC, ASC), and Speaker Recognition (SD, ER, SAR, SGR). F1-score (0-100) is the primary metric, except for ASR, where Word Accuracy Rate (WAR) is used. The Avg. column shows the mean score across available tasks. Best scores within each pruning ratio are in **bold**.

| Method | Model | Speech Content Extraction | | | Audio Classification | | | Speaker Recognition | | | | Avg. |
|---|---|---|---|---|---|---|---|---|---|---|---|---|
| | | SER | SCR | ASR | SED | MC | ASC | SD | ER | SAR | SGR | |
| *Vanilla* | | | | | | | | | | | | |
| | Phi-4-Multimodal | 18.4 | 69.3 | 92.7 | 55.1 | 46.7 | 23.4 | 26.4 | 27.3 | 26.6 | 91.1 | 47.7 |
| | Aero-1-Audio | 17.9 | 56.6 | 43.7 | 55.0 | 83.9 | 39.9 | 33.7 | 32.0 | 17.8 | 47.5 | 42.8 |
| | Qwen2.5-Omni-3B | 25.2 | 82.3 | 94.7 | 70.2 | 100.0 | 69.3 | 67.3 | 39.6 | 29.1 | 97.2 | **67.5** |
| *Light Token Pruning* (↓ **30%**) | | | | | | | | | | | | |
| Random | Phi-4-multimodal | 18.4 | 67.5 | 49.1 | 31.4 | 39.8 | 30.2 | 31.3 | 31.0 | 24.5 | 93.6 | 41.7 |
| | Aero-1-Audio | 15.9 | 53.9 | 43.3 | 56.8 | 79.4 | 40.2 | 34.0 | 32.4 | 10.0 | 38.8 | 40.5 |
| | Qwen2.5-Omni-3B | 26.5 | 80.3 | 88.4 | 71.1 | 97.5 | 69.7 | 72.0 | 38.4 | 28.6 | 95.7 | **66.8** |
| FastV | Phi-4-multimodal | 18.3 | 64.0 | 43.9 | 33.2 | 40.6 | 29.6 | 44.0 | 29.1 | 25.7 | 92.9 | 42.1 |
| | Aero-1-Audio | 19.7 | 57.0 | 37.5 | 57.0 | 78.8 | 41.0 | 42.1 | 32.2 | 9.2 | 39.2 | 41.4 |
| | Qwen2.5-Omni-3B | 18.7 | 68.2 | 76.3 | 61.3 | 98.4 | 57.2 | 38.5 | 31.1 | 17.3 | 97.5 | **56.5** |
| DART | Phi-4-multimodal | 16.8 | 67.6 | 57.2 | 54.5 | 46.1 | 31.8 | 23.1 | 28.6 | 27.1 | 91.6 | 44.4 |
| | Aero-1-Audio | 20.2 | 57.0 | 16.4 | 56.3 | 78.8 | 41.0 | 34.0 | 32.2 | 9.2 | 39.5 | 38.5 |
| | Qwen2.5-Omni-3B | 23.2 | 74.2 | 81.4 | 73.1 | 97.6 | 72.5 | 42.2 | 37.1 | 23.0 | 48.7 | **57.3** |
| Frame (Ours) | Phi-4-multimodal | 17.7 | 64.4 | 63.4 | 31.4 | 32.6 | 29.0 | 30.6 | 31.0 | 27.4 | 92.4 | 42.0 |
| | Aero-1-Audio | 15.6 | 53.7 | 43.4 | 54.3 | 82.5 | 39.8 | 34.4 | 32.1 | 8.1 | 37.3 | 40.1 |
| | Qwen2.5-Omni-3B | 26.8 | 80.9 | 92.2 | 70.5 | 98.5 | 70.2 | 65.0 | 36.4 | 31.4 | 96.7 | **66.9** |
| *Medium Token Pruning* (↓ **60%**) | | | | | | | | | | | | |
| Random | Phi-4-multimodal | 18.7 | 61.6 | 7.9 | 30.3 | 27.4 | 30.6 | 36.8 | 29.4 | 20.5 | 91.2 | 35.4 |
| | Aero-1-Audio | 12.1 | 49.7 | 34.9 | 54.6 | 78.2 | 41.3 | 42.5 | 34.5 | 8.8 | 34.0 | 39.1 |
| | Qwen2.5-Omni-3B | 24.2 | 75.3 | 59.7 | 68.7 | 95.8 | 68.3 | 66.6 | 37.9 | 27.2 | 93.5 | **61.7** |
| FastV | Phi-4-multimodal | 26.1 | 52.8 | 0.0 | 32.5 | 28.2 | 30.3 | 25.6 | 28.0 | 22.2 | 89.4 | 33.5 |
| | Aero-1-Audio | 20.3 | 54.2 | 30.4 | 58.0 | 80.2 | 44.5 | 34.2 | 33.4 | 9.1 | 34.5 | 39.9 |
| | Qwen2.5-Omni-3B | 18.0 | 63.8 | 39.2 | 60.5 | 97.5 | 57.8 | 44.0 | 28.6 | 17.1 | 95.3 | **52.2** |
| DART | Phi-4-multimodal | 18.0 | 61.1 | 23.7 | 53.9 | 44.8 | 25.4 | 26.3 | 29.4 | 24.4 | 88.0 | 39.5 |
| | Aero-1-Audio | 20.3 | 54.2 | 14.4 | 58.2 | 80.6 | 44.5 | 34.0 | 33.4 | 9.1 | 34.5 | 38.3 |
| | Qwen2.5-Omni-3B | 23.1 | 64.6 | 62.8 | 71.9 | 99.1 | 73.4 | 38.3 | 37.6 | 28.1 | 46.0 | **54.5** |
| Frame (Ours) | Phi-4-multimodal | 23.8 | 59.0 | 23.3 | 31.1 | 28.5 | 30.0 | 20.6 | 30.1 | 22.3 | 87.8 | 35.6 |
| | Aero-1-Audio | 14.0 | 51.7 | 42.5 | 56.4 | 80.9 | 41.1 | 33.0 | 34.5 | 9.1 | 33.3 | 39.9 |
| | Qwen2.5-Omni-3B | 25.6 | 75.3 | 82.2 | 69.0 | 100.0 | 68.3 | 65.7 | 38.6 | 28.3 | 91.0 | **64.4** |
| *Extreme Token Pruning* (↓ **90%**) | | | | | | | | | | | | |
| Random | Phi-4-multimodal | 18.7 | 35.3 | 0.0 | 29.3 | 20.6 | 29.6 | 41.9 | 33.4 | 11.1 | 67.5 | 28.7 |
| | Aero-1-Audio | 10.1 | 47.6 | 5.1 | 43.4 | 70.3 | 44.9 | 47.5 | 32.2 | 14.6 | 33.3 | 34.9 |
| | Qwen2.5-Omni-3B | 24.0 | 58.1 | 0.0 | 65.9 | 97.6 | 60.0 | 54.7 | 41.8 | 17.1 | 84.3 | **50.4** |
| FastV | Phi-4-multimodal | 23.4 | 43.0 | 0.0 | 27.6 | 30.1 | 29.3 | 46.3 | 24.9 | 17.3 | 82.9 | 32.5 |
| | Aero-1-Audio | 18.0 | 50.6 | 8.3 | 55.8 | 69.0 | 45.0 | 38.2 | 26.3 | 16.8 | 33.6 | 36.2 |
| | Qwen2.5-Omni-3B | 16.8 | 54.9 | 3.5 | 65.2 | 95.9 | 55.9 | 49.5 | 32.7 | 14.8 | 86.5 | **47.6** |
| DART | Phi-4-multimodal | 16.8 | 49.3 | 0.0 | 52.0 | 40.2 | 24.4 | 31.9 | 27.4 | 18.6 | 77.2 | 33.8 |
| | Aero-1-Audio | 18.0 | 50.6 | 0.0 | 55.8 | 69.0 | 45.0 | 38.1 | 26.3 | 16.8 | 33.6 | 35.3 |
| | Qwen2.5-Omni-3B | 17.3 | 54.1 | 62.9 | 66.8 | 99.1 | 69.3 | 25.6 | 42.6 | 22.1 | 52.8 | **51.3** |
| Frame (Ours) | Phi-4-multimodal | 24.6 | 36.8 | 0.0 | 28.4 | 25.0 | 28.1 | 34.8 | 30.1 | 12.1 | 66.6 | 28.7 |
| | Aero-1-Audio | 9.7 | 48.9 | 3.1 | 43.8 | 73.0 | 44.0 | 54.2 | 33.2 | 16.1 | 33.3 | 35.9 |
| | Qwen2.5-Omni-3B | 22.8 | 58.2 | 0.0 | 64.5 | 95.0 | 60.9 | 51.9 | 41.1 | 18.1 | 87.0 | **50.0** |

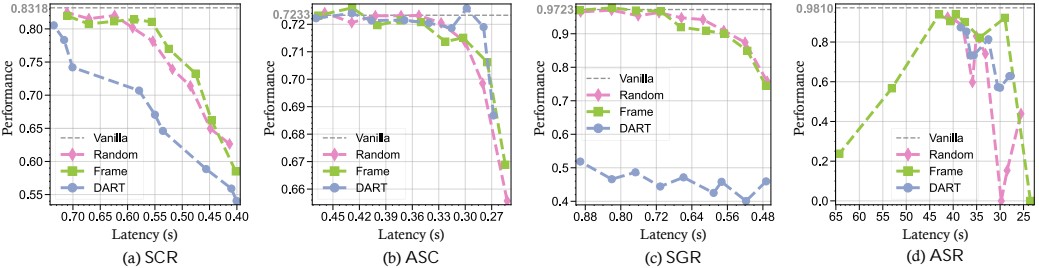

Figure 5: Comparisons of latency and performance trade-off for the Qwen2.5-Omni-3B model under different token pruning strategies across four representative datasets. Frame outperforms other methods on speech content extraction tasks across different latency constraints in almost all cases.

## 4.2 FRAME ALGORITHM

**Overview.** A training-free and time-aligned pruning method that keeps a uniformly sampled subset of audio tokens while leaving non-audio context untouched.

**Definition:** Given a mixed-modality token sequence, let $[t_0, t_0 + L)$ denote the contiguous range occupied by audio encoder tokens and let $r \in [0, 1)$ be the configured pruning ratio. The Frame operator $\mathcal{F}(t_0, L, r)$ returns the index set $\mathcal{I}_{\text{frame}}$ of tokens that remain after pruning, constructed as follows:

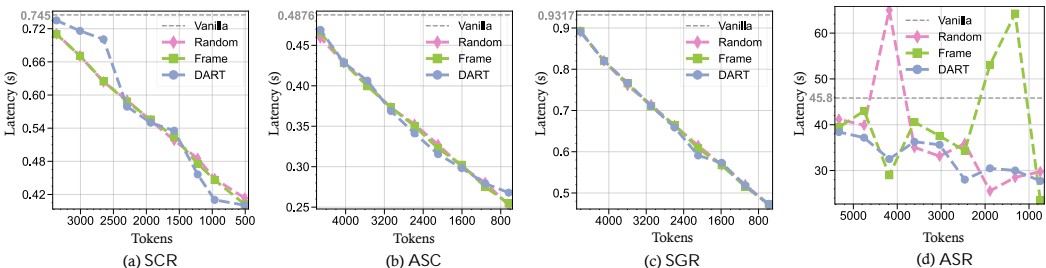

Figure 6: Acceleration effects across token pruning strategies for the Qwen2.5-Omni-3B model under various token pruning strategies across four datasets.

1. **Budget computation. The number of retained audio tokens is**

$$K = \max\Big(1, \big\lfloor L(1-r) \big\rfloor\Big), \tag{1}$$

2. **Uniform sampling.** If $K = L$ the audio span remains untouched; otherwise Frame selects indices at equal spacing

$$\mathcal{I}_{\text{audio}} = \big\{ t_0 + \lfloor j\Delta \rfloor \mid j = 0, \dots, K-1 \big\}, \tag{2}$$

where $\Delta = L/K$ is the sampling step.

3. **Index aggregation.** The final keep set simply appends the sampled audio indices to the untouched prefix and suffix tokens,

$$\mathcal{I}_{\text{frame}} = [0, t_0) \cup \mathcal{I}_{\text{audio}} \cup [t_0 + L, T), \tag{3}$$

followed by reordering in ascending order before the decoder layer is executed.

Hidden states, rotary position embeddings, and cache positions are gathered using $\mathcal{I}_{\text{frame}}$, after which the causal mask is recomputed and the decoder proceeds.

## 4.3 KV-CACHE EVICTION

We evaluate four eviction strategies under compression ratios of 30%, 60%, and 90%. The baseline, Random eviction, uniformly removes cache entries, providing a lower bound on performance under cache pressure. KNorm (Devoto et al., 2024) evicts tokens according to the L2 norm of their key vectors, based on the intuition that smaller norms contribute less to attention. TOVA (Oren et al., 2024) greedily discards tokens with minimal attention from the latest query by averaging attention weights across heads at each decoding step. Finally, SnapKV (Li et al., 2024b) retains high-attention tokens along with their neighbors using cumulative-attention scoring and 1D pooling-based clustering, preserving local semantic coherence while enabling efficient compression.

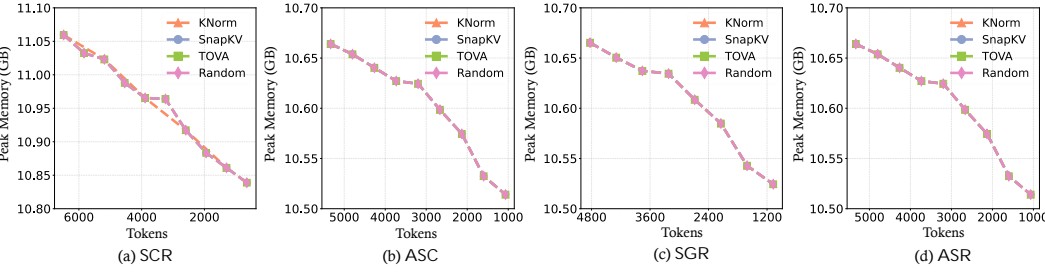

Figure 7: Peak GPU memory usage count for KV cache eviction policies applied to the Qwen2.5-Omni-3B model across four datasets, illustrating memory compression benefits during the prefilling stage for long-context audio inference.

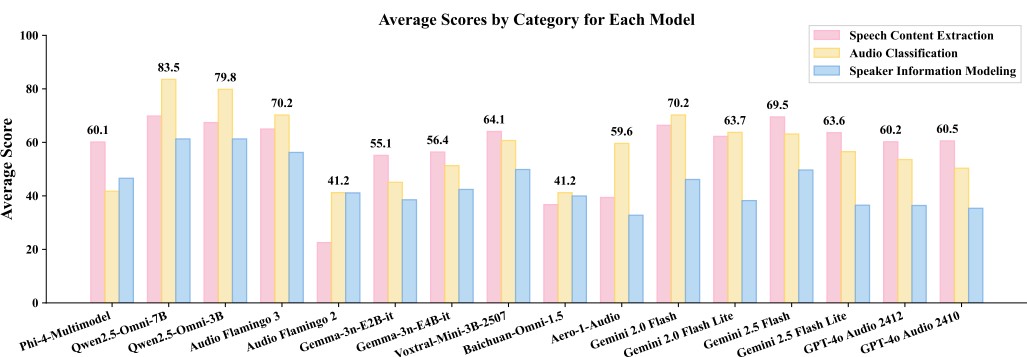

Figure 8: Average F1-scores across the three main task categories: Speech Content Extraction, Audio Classification, and Speaker Information Modeling, underscoring the need for enhanced temporal reasoning in extended audio contexts.

## 5  RESULTS AND DISCUSSIONS

Our proposed AUDIOMARATHON provides a realism-oriented evaluation framework for LALMs, focusing on minute-scale recordings, diverse audio domains, and complex reasoning. Results in Table 3 reveal clear performance stratification among the 16 evaluated models. Our key findings are summarized as follows: The best-performing model, Qwen2.5-Omni-7B, achieves an average F1-score of 70.5, whereas most open-source models cluster between 30 and 60, highlighting a substantial performance gap. By contrast, closed-source models perform unevenly: all fail on long-audio emotion recognition and authenticity detection. Only Gemini-2.5-Flash exceed 30 in emotion recognition, and all closed-source models remain below 35. To compare, human evaluation clearly surpasses the strongest model. The gap is most pronounced in speaker information modeling tasks such as speech emotion recognition (SER) and speaker-based entity recognition (ER), where human performance (87.6) remains far above model scores (generally below 65). This pronounced weakness directly reflects the challenges of entity tracking and temporal reasoning discussed in the introduction. Notably, all state-of-the-art models, both open-source and closed-source, exhibit a considerable performance deficit when compared to human competence, particularly in complex speaker information modeling tasks, underscoring the necessity to enhance entity tracking and temporal reasoning capabilities.

**Performances in area of Semantic and Acoustic.** Recent LALMs integrate acoustic and linguistic features within a single end-to-end model, enabling joint learning of cross-modal dependencies (Peng et al., 2024). In AUDIOMARATHON, ASR, Speech Content Reasoning (SCR), and Long-form Speech Entity Recognition (SER) are considered semantically sensitive tasks, while the remaining seven tasks are acoustically sensitive. Figure 8 shows that all closed-source models generally achieve around a 60 F1-score on semantically sensitive tasks. This performance reflects their baseline strength in extracting long-form speech content and performing content-based reasoning.The results on acoustically sensitive tasks exhibit significant variance. For example, the top four LALMs score above 70 on audio classification, with the strongest model reaching 83.5, indicating extensive training on classification tasks. In contrast, all models underperform significantly on speaker-related tasks, failing to exceed a 65 F1-score. In the semantic domain, the biggest challenge is Long-form SER. The top model achieves an F1-score just above 30 on SER, with only Audio-Flamingo-3 exceeding 40, indicating a substantial capability gap among current LALMs in accurately localizing and identifying entities in long-form audio. In the acoustic domain, the biggest challenge lies in speaker recognition, particularly for tasks like Speaker Age Recognition (SAR). The overall underperformance on speaker-related tasks suggests that speaker information modeling remains under-emphasized in current LALMs development. These two substantial challenges—improving long-form entity recognition and enhancing robust speaker modeling—provide crucial potential research directions for subsequent related studies.

**From Identification of Challenges to the Exploration of Solution.** Beyond long-audio understanding capabilities, memory consumption, inference latency and temporal dependencies pose additional challenges. We further explore potential mitigation strategies via token pruning and cache eviction mechanisms. The Figure 7 shows that cache eviction reduces little peak memory during prefilling, while the preserved first output token maintains for most MCQs. Figure 6 indicates that processing all

long-audio tokens at the second decoder layer is computationally expensive; reducing tokens to 10% cuts processing time to 56% of the baseline, achieving a 1.8× speedup. We find that unlike vision tokens, audio tokens encode strong temporal dependencies. Aggressive eviction can disrupt temporal coherence, especially in ASR, where every phoneme matters. Figure 5 shows that improper pruning can produce repetitive tokens, increasing latency and lowering word accuracy. Overall, task-aware audio token compression provides runtime and memory savings and is crucial for scaling long-audio LLM inference.

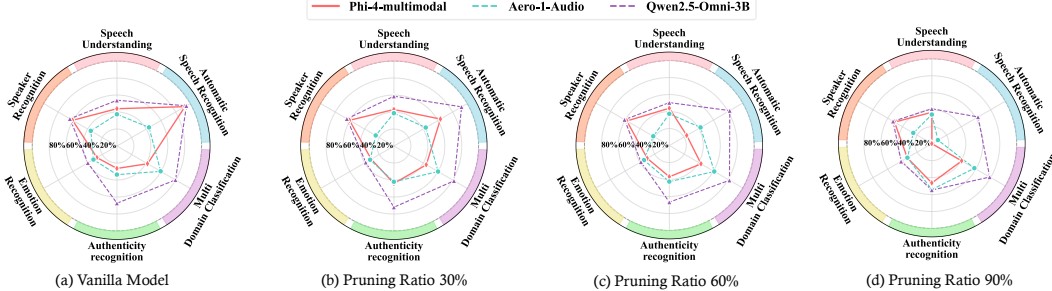

(a) Vanilla Model      (b) Pruning Ratio 30%      (c) Pruning Ratio 60%      (d) Pruning Ratio 90%

Figure 9: Performance comparison between Qwen2.5-Omni-3B, Phi-4-Mutimodal and Aero-1-Audio on six-degree capability under varying token pruning ratios.

**Redundancy Analysis in Audio Token Embeedings.** To more intuitively illustrate how token pruning affects the overall capability of the LALMs, we evaluate six-degree: (i) *Speech Understanding* (mean of SER and SCR), (ii) *Speaker Recognition* (mean of SAR and SGR), (iii) *Emotion Recognition* (ER), (iv) *Authenticity Recognition* (SD), (v) *Automatic Speech Recognition* (ASR), and (vi) *Multi-Domain Classification* (mean of SED and MC), as shown in Figure 9. For audio token pruning, the reported scores are **the maximum F1-score achieved** across tested pruning settings. **Qwen2.5-Omni-3B sustains performance** from 30% pruning, showing consistent gains across tasks, most notably in Authenticity Recognition (4.7 points). In contrast, **Aero-1-Audio** and **Phi-4-Multimodal** are highly **pruning-sensitive**. Both models suffer sharp degradation in Speech Understanding, particularly under DART or aggressive settings. While their Multi-Domain Classification performance remains relatively robust when using Frame. Due to variations in the Audio Encoders across different LALMs, pruning efficacy exhibits heterogeneity. Qwen2.5-Omni-3B demonstrates enhanced performance gains under pruning, suggesting that its original audio tokenization contains substantial uninformative redundancy, potentially even noise. Its fine-grained encoding also reveals the inherent redundancy as shown in Appendix C. In this case, pruning functions analogously to a regularization or denoising mechanism, enabling the model to concentrate more effectively on core semantic information. Conversely, performance degradation in Aero-1 and Phi-4 indicates that their token encoding is more compact, or that their architectures rely more heavily on complete sequential information, with pruning directly precipitating feature loss.

**Token-pruning effects are highly task-aware.** Table 4 highlights a task-dependent sensitivity gradient: temporally fine-grained tasks (ASR, Speech Understanding) degrade sharply when selective or attention-driven pruning removes temporally unique phonetic cues, while more global classification tasks (e.g., music detection) remain resilient even under aggressive compression. This marked disparity in sensitivity poses a substantial challenge to designing a unified approach that excels across both task categories. Therefore, we posit that future token pruning methodologies should be differentiated for these two distinct task types. To further investigate the intrinsic properties of audio tokens, we introduce the Frame strategy, aiming to examine the criticality of preserving the temporal sequentiality of token distributions through comparative analysis. Empirically, we observe that for semantic understanding tasks, Frame significantly outperforms DART, where redundancy-focused selection risks discarding rare temporal segments, and FastV, which relies on an attention-based policy. Furthermore, it demonstrates superior performance over random pruning in ASR tasks. However, we also acknowledge the unexpectedly robust performance of random pruning overall. This phenomenon not only implies a high degree of redundancy in the audio tokenization of current LALMs but also underscores the necessity for developing more efficient and theoretically sound audio token pruning methodologies to fill this disciplinary gap, rather than directly migrating methods from vision or text domains.

## 6 RELATED WORKS

**Large Audio Language Models.** The development of LALMs follows the broader shift toward multimodal language processing. Early systems combined ASR and Text-to-Speech with text-based LLMs for audio-to-text tasks, but they suffered from error propagation and weak cross-modal fusion (Ngiam et al., 2011; Hinton et al., 2012; Wang et al., 2017). Self supervised speech representations, such as wav2vec 2.0 (Baevski et al., 2020) and HuBERT (Hsu et al., 2021), drove major progress and enabled models like Whisper (Radford et al., 2023) and SpeechGPT (Zhang et al., 2023a). Recent instruction-tuned Audio LLMs, such as Phi-4-multimodal (Abouelenin et al., 2025), Freeze-Omni (Wang et al., 2024b), and Qwen2.5-Omni (Xu et al., 2025a), unify audio and language within one framework and support tasks that span ASR, audio question answering, and audio understanding. As context windows grow (Liu et al., 2025), these models are also moving toward longer audio inputs, with some reporting support for hours of audio.

**Audio LLM Benchmarks.** Benchmarking has evolved from task-specific datasets to broader frameworks that test multimodal and instruction-following abilities. Early datasets such as AudioSet (Gemmeke et al., 2017), LibriSpeech (Panayotov et al., 2015), ESC 50 (Piczak, 2015), and FSD50K (Fonseca et al., 2021) focused on classification or ASR. SUPERB (Yang et al., 2021) expanded speech evaluation with a broader task set. For audio language understanding, Clotho QA (Drossos et al., 2020) and AudioCaps (Kim et al., 2019) introduced question answering and captioning. More recent datasets, such as MMAU (Sakshi et al., 2024) and AIR Bench (Yang et al., 2024), target instruction following and tri modal reasoning. Despite progress, few benchmarks directly test long audio comprehension or the efficiency of long sequence processing.

**Token Compression.** Transformer-based models face memory and compute limits with long context and multimodal inputs. Two practical directions are KV cache eviction and token pruning (Liu et al., 2025; Wen et al., 2025a; Xiong et al., 2025; Yang et al., 2025; Chen et al., 2025). For KV cache eviction, SnapKV (Li et al., 2024b) clusters high attention tokens and stores centroids, H2O (Zhang et al., 2023b) balances recent and salient tokens, and StreamingLLM (Xiao et al., 2023) uses fixed attention sinks and a sliding window for unbounded generation. In vision language models, token pruning reduces redundant visual tokens through architectural methods, such as Q-Former context tokens (Li et al., 2024a), and through inference time methods, such as Token Merging (Bolya et al., 2022), FastV (Chen et al., 2024a), SparseVLM (Zhang et al., 2024b), and DART (Wen et al., 2025b). While these methods are effective for vision or text tokens, research on audio token compression remains limited, and it is unknown how well these methods transfer to the audio modality.

## 7 CONCLUSION

We present AUDIOMARATHON, a comprehensive benchmark for LALMs that targets minute-scale speech, sound, and music inputs, spanning 10 representative audio tasks. Through these long-form scenarios, AUDIOMARATHON exposes fundamental challenges such as long-range dependency, temporal continuity, and source confusion. While existing LALMs perform well on short-range tasks like classification and reasoning, they struggle with long-span speech understanding and speaker analysis, revealing limitations in consistency and entity tracking, which exhibit a significant gap compared to human performance, providing a potential direction for future research. Moreover, the field of large audio models lacks attention to the efficiency of audio encoders, which, in practice, leads to substantial redundancy in audio tokens. Ultimately, AUDIOMARATHON provides a foundation for developing robust and efficient long-audio modeling.

## ETHICS STATEMENT

Our research introduces the AUDIOMARATHON to advance long-form audio understanding and inference efficiency in Large Audio Language Models (LALMs). We acknowledge the dual-use potential of this technology, which could be misused for generating deepfake audio or eroding privacy. We justify its public disclosure as a means to foster robustness and safety through transparent benchmarking and to highlight model limitations for proactive risk mitigation. We encourage future work to expand this effort with responsible practices across diverse languages and contexts. We hereby affirm that this work was conducted in strict compliance with academic ethics, with the primary goal of steering technological progress toward beneficial ends; any misuse of this research for unlawful or unethical purposes is unequivocally contrary to our principles.

## REPRODUCIBILITY STATEMENT

We are committed to ensuring the reproducibility of our results. To this end, we provide comprehensive details about our experimental setup and datasets. Specifically, all hyperparameters, model descriptions, pruning strategies, and evaluation protocols are specified in the main text. Additional analysis, including random baseline results, error analysis, encoding granularity of LALMs, and model details, is presented in the Appendix E. We describe all datasets used in our benchmark (AUDIOMARATHON) in Appendix D, including their construction and task definitions. Detailed motivation and implementation of pruning at the second layer are provided in Appendix A, where we explain its design as an early-stage compression mechanism for audio tokens. We report the performance of different pruning methods, random baselines, acceleration consistency, and model-specific results (e.g., Qwen2.5-Omni-3B) in the appendix for full transparency. Appendix B provides descriptions of all baseline models, including their architectures, training strategies, and modality support. Together, these efforts ensure that all experiments can be independently reproduced.

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

APPENDIX

## A  ADDITIONAL RESULTS

**Token Pruning Details.** In multi-modal
large language models, the first two layers
are regarded as shallow layers, where atten-
tion allocation remains relatively balanced,
and output tokens mainly attend to preced-
ing outputs while modality-specific tokens
(e.g., vision or audio) are not yet fully in-
tegrated into semantic reasoning. Prior
analysis on vision tokens has demonstrated
that pruning at the second layer is particu-
larly effective: it removes redundant tokens
while retaining a compact set of representa-
tive ones, thereby preventing redundant
information from propagating into deeper
layers and significantly reducing computa-
tional overhead. Motivated by this observa-
tion, we apply the same strategy to audio
tokens, pruning them directly at the sec-
ond layer. This early pruning leverages the
redundancy of low-level acoustic represen-
tations, which often contain overlapping
information, and achieves a favorable bal-
ance between efficiency and performance.
Compared with pruning at the first layer,

Table 5: Core statistics of the AudioMarathon.

| Statistics | Number |
|---|---|
| Total Questions | 6567 |
| Audio Domains | 10 |
| Difficulty (Easy:Medium:Hard) | 24%:61%:15% |
| Speech Content Extraction | 1514 |
| Automatic Speech Recognition (ASR) | 204 (3.10%) |
| Speech Content Reasoning (SCR) | 820 (12.49%) |
| Speech Entity Recognition (SER) | 490 (7.46%) |
| Audio classification | 1519 |
| Audio scene classifier (ASC) | 1145 (17.44%) |
| Music classifier (MC) | 120 (1.83%) |
| Sound event detection (SED) | 254 (3.87%) |
| Speaker Recognition | 3530 |
| Emotion Recognition (ER) | 185 (2.82%) |
| Speech Detection (SD) | 776 (11.82%) |
| Speaker Age Recognition (SAR) | 959 (14.60%) |
| Speaker Gender Recognition (SGR) | 1614 (24.58%) |
| Mutiple Choice Questions | 6452 |
| Transcriptions | 270 |

where feature representations are still un-

stable and critical information may be lost, the second layer offers a more appropriate trade-off between stability and efficiency. Conversely, deferring pruning to deeper layers would result in repeated computations on redundant tokens, diminishing overall efficiency. Thus, second-layer pruning of audio tokens can be understood as a low-level information compression mechanism, which eliminates ineffective tokens at an early stage to maximize acceleration in subsequent layers while maintaining robust performance.

## A.1 RANDOM CHOICE BASELINE

Table 6: Random baseline results on nine subsets of the dataset after 100 random selections. General refers to double-choice questions; Four-choice refers to Four-option choice questions.

| Task | Labels | Type | ACC | Macro F1-score |
|------|--------|------|-----|----------------|
| MC | 4 | four-choice | 0.2477 | 0.2451 |
| SD | 2 | general | 0.5200 | 0.4700 |
| SER | 4 | four-choice | 0.2525 | 0.2518 |
| ASC | 4 | four-choice | 0.2496 | 0.2493 |
| ER | 4 | four-choice | 0.2436 | 0.2426 |
| SGR | 2 | general | 0.5000 | 0.4798 |
| SAR | 5 | general | 0.1979 | 0.1695 |
| SCR | 4 | four-choice | 0.2490 | 0.2491 |
| ASC | 10 | general | 0.2467 | 0.2450 |
| Four-choice (6 tasks) | | | 0.2490 | 0.2479 |
| General (3 tasks) | | | 0.3999 | 0.3744 |
| Overall (9 tasks) | | | 0.2993 | 0.2901 |

The Table 6 illustrates that the F1-scores are lower, directly reflecting the validity of using F1-score as an evaluation metric, highlighting its ability to balance precision and recall.

## A.2 TOKEN PRUNING RESULTS OF QWEN2.5-OMNI-3B ON THE OTHER SIX DATASETS IN AUDIOMARATHON

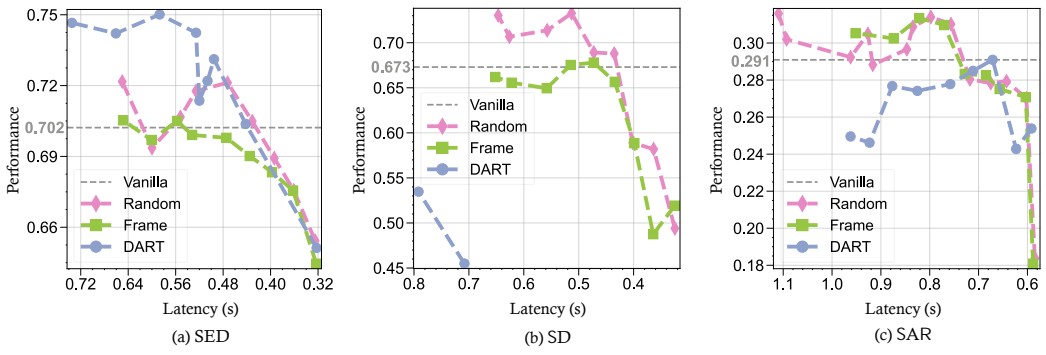

(a) SED  (b) SD  (c) SAR

Figure 10: Comparisons of latency and performance trade-off for the Qwen2.5-Omni-3B model on the SED, SD, and SAR dataset.

While prior results demonstrate the robustness and general superiority of the Frame method on representative MCQs, we further investigate the specific advantages and disadvantages exhibited by certain methods on particular tasks. For instance, on simpler tasks like SED and MC, Frame performs stably, surpassing random methods. However, it underperforms on more challenging tasks, such as SER and SAR. In contrast, Frame maintains robust performance, proving its relative reliability.

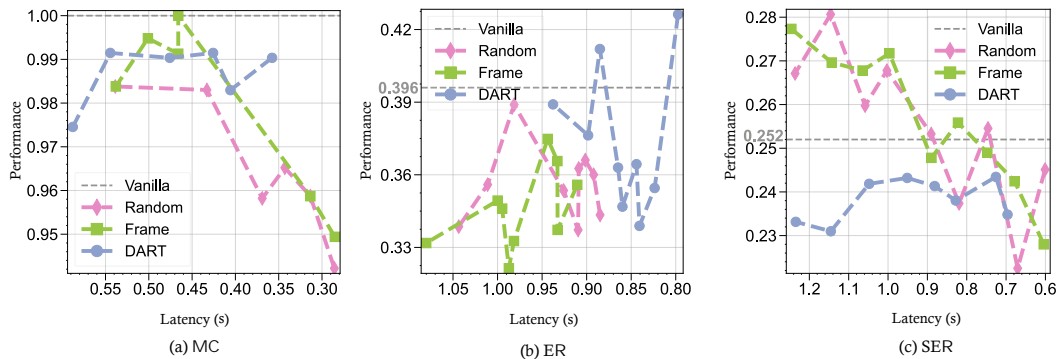

Figure 11: Comparisons of latency and performance trade-off for the Qwen2.5-Omni-3B model on the MC, ER, and SER dataset.

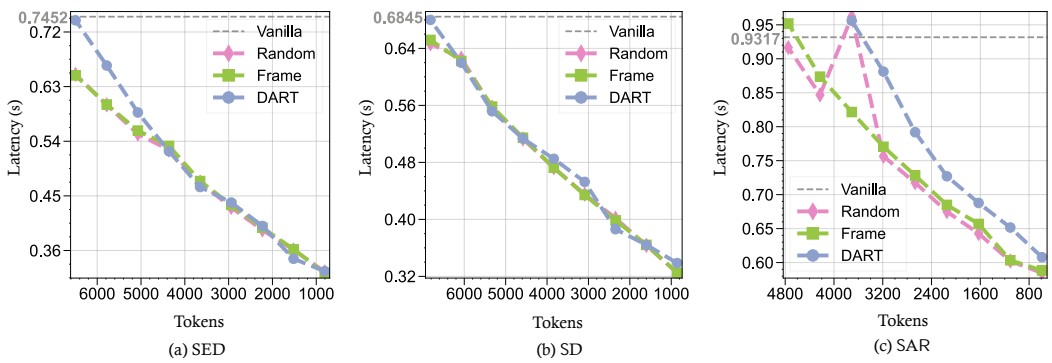

Figure 12: Acceleration effects for the Qwen2.5-Omni-3B model on the SED, SD, and SAR dataset.

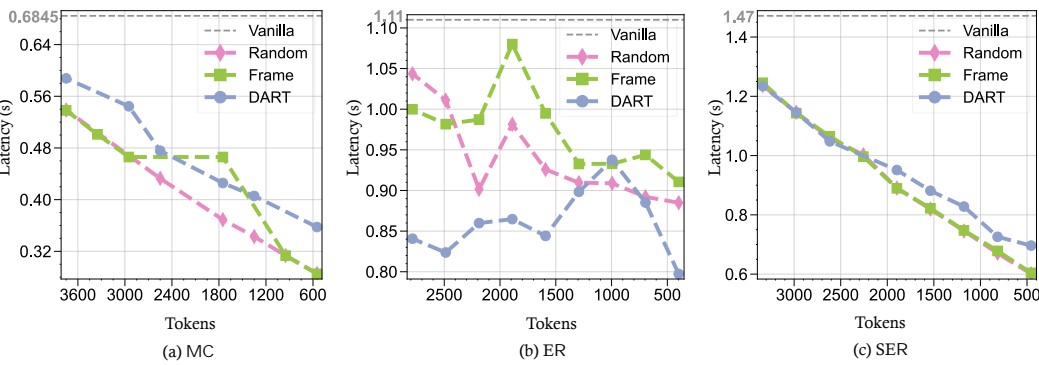

Figure 13: Acceleration effects for the Qwen2.5-Omni-3B model on the MC, ER, and SER dataset.

As shown in Figure 12 and Figure 13, different token pruning methods maintain a high degree of consistency in acceleration performance, further illustrating their effectiveness in reducing inference time for long audio MCQ tasks.

## A.3 CACHE EVICTION RESULTS OF QWEN2.5-OMNI-3B IN AUDIOMARATHON

Table 7: Performance comparison of KVPress pruning methods on AudioMarathon tasks, grouped into Speech Content Extraction (SER, SCR, ASR), Audio Classification (SED, MC, ASC), and Speaker Information Modeling (SD, ER, SAR, SGR). Macro F1-scores (0–100) are reported.

| Method | Ratio | Speech Content Extraction | | | Audio Classification | | | Speaker Info. Modeling | | | |
|---|---|---|---|---|---|---|---|---|---|---|---|
| | | SER | SCR | ASR | SED | MC | ASC | SD | ER | SAR | SGR |
| Vanilla | 0.0 | 25.2 | 82.3 | 94.7 | 70.2 | 97.4 | 69.3 | 67.3 | 39.6 | 29.1 | 97.2 |
| KNORM | 0.3 | 25.2 | 82.3 | 91.2 | 70.2 | 97.4 | 69.3 | 67.3 | 39.6 | 29.1 | 97.2 |
| | 0.5 | 25.2 | 82.3 | 35.7 | 70.2 | 97.4 | 69.3 | 67.3 | 39.6 | 29.1 | 97.2 |
| | 0.6 | 25.2 | 82.3 | 4.1 | 70.2 | 97.4 | 69.3 | 67.1 | 39.6 | 29.1 | 97.2 |
| | 0.7 | 25.2 | 82.2 | 0.0 | 70.1 | 97.4 | 69.2 | 66.8 | 39.6 | 28.9 | 97.0 |
| | 0.9 | 24.9 | 82.0 | 0.0 | 70.0 | 96.8 | 68.9 | 66.8 | 38.3 | 28.6 | 97.0 |
| RANDOM | 0.3 | 25.2 | 82.3 | 77.0 | 70.2 | 97.4 | 69.3 | 67.3 | 39.6 | 29.1 | 97.2 |
| | 0.5 | 25.2 | 82.3 | 21.4 | 70.2 | 97.4 | 69.3 | 67.3 | 39.6 | 29.1 | 96.9 |
| | 0.6 | 25.2 | 82.3 | 0.0 | 70.2 | 97.4 | 69.3 | 67.3 | 39.6 | 29.1 | 97.2 |
| | 0.7 | 25.2 | 82.3 | 0.0 | 69.8 | 98.1 | 69.0 | 66.7 | 39.2 | 28.7 | 96.6 |
| | 0.9 | 25.4 | 82.1 | 0.0 | 69.6 | 97.9 | 69.0 | 66.3 | 38.9 | 27.5 | 96.4 |
| SNAP | 0.3 | 25.2 | 82.3 | 32.8 | 70.2 | 97.4 | 69.3 | 67.3 | 39.6 | 29.1 | 97.2 |
| | 0.5 | 25.0 | 82.3 | 0.0 | 70.2 | 97.4 | 69.3 | 67.3 | 39.6 | 29.1 | 97.2 |
| | 0.6 | 25.2 | 82.3 | 0.0 | 70.2 | 97.4 | 69.3 | 67.3 | 39.6 | 29.1 | 97.2 |
| | 0.7 | 25.2 | 81.9 | 0.0 | 70.1 | 97.4 | 68.8 | 67.3 | 39.1 | 29.1 | 96.9 |
| | 0.9 | 25.0 | 81.8 | 0.0 | 69.9 | 96.8 | 68.6 | 67.0 | 39.0 | 26.5 | 97.2 |
| TOVA | 0.3 | 25.2 | 82.3 | 88.6 | 70.2 | 97.4 | 69.3 | 67.3 | 39.6 | 29.1 | 97.2 |
| | 0.5 | 25.0 | 82.3 | 42.5 | 70.2 | 97.4 | 69.3 | 67.3 | 39.6 | 29.1 | 97.2 |
| | 0.6 | 25.2 | 82.3 | 11.1 | 70.2 | 96.2 | 69.3 | 66.8 | 39.6 | 29.1 | 97.2 |
| | 0.7 | 25.2 | 82.1 | 0.0 | 69.8 | 97.4 | 68.6 | 66.5 | 38.2 | 26.8 | 96.8 |
| | 0.9 | 24.9 | 81.9 | 0.0 | 69.0 | 96.3 | 68.3 | 66.2 | 37.8 | 24.5 | 96.8 |

From our cache-eviction experiments, we observe that applying KVPress compression in the prefilling stage has virtually low impact performance for QA-style tasks. These findings align with prior work: KV-Compress's LongBench results show that eviction-based KV compression attains near–full-cache performance on most single- and multi-document QA benchmarks even under aggressive compression rates (Rehg, 2024), and similar trends have been independently reported in Dynamic-LLaVA (Huang et al., 2024) on MLLMs. Theoretically, once the initial prefilling is complete, performing KVPress compression on the populated KV cache does not alter the model's computed logits for the first decoded token. This is because the first-token distribution remains unchanged, any perturbations introduced by compression are unlikely to produce substantial divergence in typical QA settings and therefore have only a negligible effect on overall performance. In contrast, for ASR-type tasks—where the decoding horizon is much longer and many more tokens are produced—small errors introduced by cache eviction can compound across successive decoding steps, yielding a clearly observable degradation in performance.

## A.4 ABLATION RESULTS ON SHORT AUDIO DURATION IN AUDIOMARATHON

Table 8: Impact of Audio Duration on Model Performance. Comparison between short audio segments and long-form audio. **Bold** numbers indicate the better performance between the two duration settings for each model.

| Model | Duration | Speech Content Extraction | | | Audio Classification | | | Speaker Recognition | | | | Avg. |
|---|---|---|---|---|---|---|---|---|---|---|---|---|
| | | SER | SCR | ASR | SED | MC | ASC | SD | ER | SAR | SGR | |
| Qwen2.5-Omni-3B | Long (Main) | 25.2 | 82.3 | 94.7 | **70.2** | **97.4** | **69.3** | **67.3** | 39.6 | 29.1 | **97.2** | **67.2** |
| | Short (<30s) | **38.6** | **89.1** | **96.9** | 66.2 | 97.3 | 58.0 | 33.2 | **40.2** | **29.3** | 92.6 | 64.1 |
| Phi-4-Multimodal | Long (Main) | **18.4** | 69.3 | 92.7 | **55.1** | 46.7 | 23.4 | **26.4** | 27.3 | **26.6** | 91.1 | **47.7** |
| | Short (<30s) | 14.0 | **75.8** | **96.8** | 48.4 | **61.8** | **29.2** | 2.5 | 27.3 | 23.8 | 91.1 | 47.1 |
| Aero-1-Audio | Long (Main) | **17.9** | 56.6 | 43.7 | **55.0** | **83.9** | 39.9 | 33.7 | **32.0** | **17.8** | **47.5** | 42.8 |
| | Short (<30s) | 13.8 | **63.2** | **97.6** | 51.1 | 73.8 | **42.8** | **42.7** | 12.5 | 11.1 | 33.6 | **44.2** |

This result demonstrates:

**Distraction in Entity Tracking.** In the short-audio setting, Qwen2.5-Omni-3B exhibited a significant improvement in SER, rising from 25.2 in the long-audio context to 38.6, and in SCR, increasing from 82.3 to 89.1. This indicates that redundant information inherent in long audio sequences tends to dilute critical cues, causing the model to suffer from "attentional drift" when attempting to localize specific speakers or details. Conversely, the "high signal-to-noise ratio" characteristic of short-audio environments significantly mitigates the complexity of reasoning.

**Necessity for Global Discrimination.** Conversely, for tasks reliant on global statistical features, long audio demonstrates an indispensable advantage. When restricted to short audio, Qwen2.5-Omni-3B saw its SD score plummet to 33.2 (down from 67.3) and its ASC (Audio Scene Classification) score decline to 58.0 (from 69.3). This demonstrates that deepfake detection and complex scene understanding require the accumulation of acoustic evidence over extended temporal dimensions, whereas short temporal slices result in feature insufficiency.

**Stratification in Robustness.** For the lightweight model Aero-1-Audio, the short-audio environment facilitated a dramatic recovery in ASR performance, bounding from a catastrophic failure in the long-audio setting (43.7) to 97.6. This indicates severe bottlenecks in positional encoding or memory retention within its long-context encoding mechanism. In contrast, Qwen2.5-Omni-3B maintained high ASR performance across both long and short contexts (94.7 vs. 96.9), exhibiting superior robustness in processing long sequences.

# B  ERROR ANALYSIS

Table 9: Summary of Error Analysis for Qwen2.5-Omni on LibriSpeech ASR.

| Method | Error Patterns |
|--------|----------------|
| **DART 10%** | *Insertions*: 'the': 18, 'a': 15, 'main': 9, 'britain': 8, 'billy': 8, 'i': 7, 'and': 7, 'any': 5
*Deletions*: 'and': 62, 'was': 62, 'the': 61, 'it': 46, 'of': 44, 'to': 41, 'a': 38, 'one': 32
*Substitutions*: 'an' → 'and': 25, 'the' → 'a': 18, 'saint' → 'st': 13, 'a' → 'the': 13, 'and' → 'in': 12, 'this' → 'the': 11, 'every' → 'everyone': 11 |
| **DART 30%** | *Insertions*: 'celebrated': 72, 'costume': 63, 'the': 47, 'representing': 42, 'spring': 33, 'and': 13, 'a': 11, 'main': 9
*Deletions*: 'the': 486, 'and': 318, 'of': 301, 'to': 219, 'a': 206, 'was': 157, 'in': 136, 'it': 117
*Substitutions*: 'the' → 'kip': 31, 'an' → 'and': 20, 'a' → 'the': 19, 'the' → 'a': 19, 'and' → 'in': 13, 'and' → 'kip': 13, 'this' → 'the': 12 |
| **DART 50%** | *Insertions*: 'the': 74, 'different': 33, 'parts': 29, 'building': 27, 'carpenters': 19, 'a': 17, 'had': 16, 'connected': 16
*Deletions*: 'the': 1152, 'of': 675, 'and': 591, 'to': 473, 'a': 391, 'in': 333, 'that': 257, 'i': 217
*Substitutions*: 'an' → 'and': 23, 'the' → 'a': 17, 'in' → 'and': 16, 'a' → 'the': 14, 'this' → 'the': 14, 'saint' → 'st': 12, 'and' → 'in': 12, 'o' → 'of': 10 |
| **Frame 10%** | *Insertions*: 'the': 17, 'and': 10, 'a': 10, 'main': 9, 'britain': 8, 'billy': 7, 'some': 6, 'i': 6
*Deletions*: 'the': 430, 'and': 254, 'of': 177, 'a': 156, 'to': 129, 'in': 129, 'that': 96, 'her': 94
*Substitutions*: 'an' → 'and': 22, 'the' → 'a': 20, 'saint' → 'st': 16, 'a' → 'the': 13, 'and' → 'in': 11, 'every' → 'everyone': 11, 'round' → 'around': 10 |
| **Frame 30%** | *Insertions*: 'savage': 124, 'and': 19, 'the': 19, 'a': 14, 'main': 8, 'in': 7, 'britain': 7, 'it': 6
*Deletions*: 'the': 756, 'and': 375, 'of': 266, 'a': 250, 'to': 223, 'in': 182, 'that': 129, 'it': 113
*Substitutions*: 'the' → 'a': 38, 'an' → 'and': 23, 'a' → 'the': 21, 'rodolfo' → 'rodolpho': 15, 'and' → 'in': 13, 'round' → 'around': 12, 'this' → 'the': 11 |
| **Frame 50%** | *Insertions*: 'the': 28, 'and': 27, 'a': 24, 'to': 14, 'i': 12, 'main': 12, 'in': 10, 'he': 10
*Deletions*: 'the': 996, 'and': 506, 'of': 377, 'a': 332, 'to': 285, 'in': 264, 'that': 180, 'it': 177
*Substitutions*: 'the' → 'a': 51, 'a' → 'the': 28, 'an' → 'and': 26, 'this' → 'the': 21, 'saint' → 'st': 16, 'mainhall' → 'hall': 15, 'and' → 'in': 14 |
| **Random 10%** | *Insertions*: 'the': 20, 'a': 15, 'and': 9, 'main': 9, 'i': 8, 'battle': 8, 'billy': 7, 'some': 6
*Deletions*: 'the': 419, 'and': 250, 'of': 135, 'a': 127, 'to': 120, 'in': 110, 'i': 93, 'that': 73
*Substitutions*: 'an' → 'and': 22, 'the' → 'a': 18, 'a' → 'the': 14, 'saint' → 'st': 13, 'this' → 'the': 11, 'every' → 'everyone': 11, 'and' → 'in': 10 |
| **Random 30%** | *Insertions*: 'a': 72, 'long': 62, 'was': 58, 'there': 57, 'silence': 57, 'ensued': 54, 'the': 20, 'and': 14
*Deletions*: 'the': 919, 'and': 584, 'of': 335, 'to': 311, 'a': 307, 'in': 232, 'that': 193, 'he': 182
*Substitutions*: 'the' → 'a': 38, 'this' → 'the': 25, 'a' → 'the': 23, 'an' → 'and': 22, 'saint' → 'st': 17, 'and' → 'in': 17, 'his' → 'the': 13 |
| **Random 50%** | *Insertions*: 'the': 267, 'a': 247, 'thousand': 170, 'pieces': 169, 'of': 116, 'roof': 91, 'as': 83, 'author': 79
*Deletions*: 'the': 1852, 'and': 1165, 'of': 867, 'to': 669, 'a': 640, 'in': 500, 'that': 432, 'i': 386
*Substitutions*: 'the' → 'a': 77, 'a' → 'the': 31, 'this' → 'the': 29, 'in' → 'and': 22, 'an' → 'and': 21, 'his' → 'the': 20, 'to' → 'of': 16 |

Table 10: Summary of Error Analysis for Phi-4-multimodal on ASR Tasks.

| Method | Error Patterns |
|---|---|
| **Vanilla** | *Insertions*: 'to': 12, 'the': 7, 'a': 4, 'hundred': 3, 'is': 3, 'g.': 3, 'or': 3, 'be': 3, 'with': 3 |
| | *Deletions*: 'the': 13, 'that': 5, 'a': 4, 'percent': 4 |
| | *Substitutions*: 'the' → 'a': 6, 'in' → 'and': 3, 'had' → 'has': 3, 'it' → 'it's': 3, 'a' → 'the': 3 |
| **FastV Prune 20%** | *Insertions*: 'to': 11, 'the': 8, 'hundred': 4, 'or': 4, 'be': 4, 'and': 3 |
| | *Deletions*: 'the': 451, 'and': 152, 'to': 139, 'of': 126, 'is': 122, 'in': 77, 'a': 76, 'that': 45, 'was': 43, 'are': 40 |
| | *Substitutions*: 'the' → 'a': 7, 'in' → 'and': 3, 'is' → 'the': 3, 'a' → 'the': 3 |
| **Dart Prune 20%** | *Insertions*: 'to': 12, 'the': 7, 'a': 5, 'is': 4, 'g.': 3, 'or': 3, 'be': 3, 'with': 3 |
| | *Deletions*: 'the': 52, 'to': 30, 'is': 26, 'are': 13, 'a': 9, 'and': 9, 'that': 8, 'of': 7, 'was': 7, 'in': 6 |
| | *Substitutions*: 'the' → 'a': 6, 'a' → 'the': 4, 'had' → 'has': 3 |

In this section, we further compare the token prune results on ASR task. The result of Table 9 and Table 10 demonstrates the attention-based selection probably causes the loss of high-frequency words. The model's substantial omission of high-frequency words in the audio transcription task results in inferior performance under the same pruning ratio.

## C ENCODING GRANULARITY OF LALMS

Table 11: Audio processing capacity of Audio Language Models, including maximum supported audio length, maximum number of encoded audio tokens, and token rate (tokens per second).

| Model Name | Max Audio Length | Max Encoded Audio Tokens | Token Rate (tokens/s) |
|---|---|---|---|
| Phi-4-multimodal (Abouelenin et al., 2025) | 30 minutes | 22500 | 12.5 tokens/s |
| Aero-1-Audio (Li et al., 2025a) | 15 minutes | 22500 | 25.0 tokens/s |
| Qwen2-Audio-Instruct (Chu et al., 2024) | 0.5 minutes | 750 | 25.0 tokens/s |
| Qwen2.5-Omni (Xu et al., 2025a) | 21 minutes | 32000 | 25.0 tokens/s |
| Qwen3-Omni (Xu et al., 2025b) | 40 minutes | 32,768 | 12.5 tokens/s |
| Step-Audio 2 (Wu et al., 2025) | 10 minutes | 8000 (with text) | 12.5 tokens/s |

Table 11 reports the audio encoding granularity of the LALMs. Except for Phi-4-Multimodal, all models produce about 7,500 tokens for a 5-minute clip, even for straightforward tasks such as gender or age classification, which reveals substantial redundancy in current audio encoding.

Notably, recent leading LALMs have already started to explicitly address the redundancy of audio token embeddings at the architectural level. For example, in the audio encoding design of Qwen3-Omni, the dedicated audio encoder increases the temporal span represented by each token from 40 ms in Qwen2.5-Omni to 80 ms (Xu et al., 2025b), effectively halving the token rate. Similarly, Step-Audio 2, another strong LALM, employs an audio adaptor with a downsampling rate of 2 to connect the audio encoder to the LLM, thereby reducing the output frame rate of the audio encoder to 12.5 Hz, which equals to 80 ms per token (Wu et al., 2025). These designs indicate that teams of state-of-the-art LALMs have become aware of the redundancy in audio token embeddings and are actively exploring more compact audio representations.

## D  DATASET CONSTITUTE

**SLUE** (Shon et al., 2022). The Spoken Language Understanding Evaluation (SLUE) benchmark is a suite of tasks designed for evaluating speech models on spoken language understanding. It is derived from the full 960 hours of the LibriSpeech corpus and includes various tasks such as named entity recognition (NER), sentiment analysis, and relation extraction. For AUDIOMARATHON, we utilize the sentiment analysis subset, which requires models to comprehend spoken content and infer the underlying sentiment.

**RACE** (Lai et al., 2017). The Reading Comprehension from Examinations (RACE) dataset is a large-scale collection of reading comprehension questions from English exams for middle and high-school Chinese students. It consists of over 28,000 passages and nearly 100,000 questions written by human experts to evaluate reading comprehension and reasoning skills. In AUDIOMARATHON, we use an audio-transcribed version of the RACE dataset, transforming the text-based reasoning challenge into a listening comprehension task that tests a model's ability to process and reason over long spoken narratives.

**LibriSpeech-long** (Park et al., 2024). LibriSpeech is a widely used corpus for Automatic Speech Recognition (ASR), containing approximately 1,000 hours of English speech read from public domain audiobooks. The original dataset consists of short audio clips, typically a few seconds long. For AUDIOMARATHON, we created LibriSpeech-long by concatenating multiple short clips from the same speaker and chapter to form continuous, long-form audio files, which are used to evaluate the models' long-context ASR performance.

**DESED** (Turpault et al., 2019). The Domestic Environment Sound Event Detection (DESED) dataset is designed for the task of sound event detection in real-life domestic environments. The dataset is built using a combination of synthesized and real recordings from AudioSet, focusing on 10 common domestic sound classes (e.g., dog bark, blender, speech). It provides strong annotations with precise event start and end times, making it a challenging benchmark for evaluating the temporal localization capabilities of audio models.

**GTZAN** (Tzanetakis & Cook, 2002). GTZAN Genre Collection is one of the most widely used datasets for music genre classification. It consists of 1,000 audio tracks, each 30 seconds long, distributed evenly across 10 distinct music genres (e.g., Blues, Classical, Hip-Hop, Jazz, Rock). Each genre is represented by 100 clips. Despite some known issues with label consistency in a small fraction of the data, it remains a standard benchmark for evaluating music information retrieval.

**TAU Urban Acoustic Scenes** (Heittola et al., 2019). TAU Urban Acoustic Scenes dataset is a collection of recordings from various acoustic scenes for the task of acoustic scene classification. The 2019 version, which we reference, contains over 40 hours of audio recorded in 10 different European cities. The data is provided as 10-second segments extracted from longer original recordings, capturing diverse urban environments such as airports, public parks, and metro stations. In our benchmark, we utilize these longer source recordings to test scene classification in extended audio contexts.

**HAD** (Yi et al., 2021). The Hallym Aging Diacrisis (HAD) dataset is a Korean speech corpus designed for the study of age-related voice characteristics and the diagnosis of pathological voices in the elderly. It contains speech samples from different age groups, including young adults and elderly individuals, performing various speech tasks like reading passages and sustained vowel phonations. The dataset is annotated with speaker age and health status, making it suitable for tasks related to age estimation and vocal health detection.

**VESUS** (Sager et al., 2019). The Voice Evaluation for Specific UtteranceS (VESUS) dataset is a corpus for assessing voice pathologies. It contains recordings from speakers with various voice disorders as well as healthy controls. Speakers were recorded producing specific utterances, such as sustained vowels and standard sentences, which are designed to highlight vocal impairments. The dataset is annotated by expert clinicians with labels for overall voice quality and specific perceptual ratings (e.g., roughness, breathiness, strain), serving as a benchmark for automated voice quality assessment systems.

**Vox_Age & Vox_Gender** (Hechmi et al., 2021). These tasks are derived from the large-scale VoxCeleb speaker recognition dataset (Nagrani et al., 2017). VoxCeleb consists of hundreds of

thousands of "in-the-wild" speech clips extracted from celebrity interview videos on YouTube. While the primary purpose of VoxCeleb is speaker identification and verification, the metadata associated with each celebrity allows for the creation of auxiliary tasks. For AUDIOMARATHON, we use this data to evaluate speaker characteristic identification, specifically age estimation (VoxAge) and gender classification (VoxGender) from long, unconstrained speech segments.

## E  MODEL DETAILS

**Phi-4-Multimodal** (Abouelenin et al., 2025). This model is extended from Phi-4-Mini and integrates three input modalities: text, vision, and speech/audio. Its key innovation lies in the use of the "Mixture-of-LoRAs" technique: while keeping the base language model completely frozen, it introduces modality-specific LoRA adapters and a routing mechanism to enable flexible multimodal reasoning (e.g., vision + language, vision + speech, speech-only) without interference across modalities.

**Qwen2.5-Omni** (Xu et al., 2025a). Developed by the Qwen Team, this is an end-to-end multimodal model capable of perceiving multiple modalities, including text, image, audio, and video, and supporting streaming generation of both text and natural speech responses. Its main innovations include: the introduction of TMRoPE (temporally aligned multimodal rotary position embedding) for audio-video timestamp synchronization; the Thinker–Talker architecture, where the Thinker is responsible for text generation and the Talker generates audio tokens based on the hidden states of the Thinker, thereby avoiding interference between text and speech generation; and the use of block-level processing and sliding-window DiT mechanisms to reduce streaming latency.

**Audio-Flamingo-2 (AF2)** (Ghosh et al., 2025). This model is an audio language model (ALM) with long audio understanding ability (30 seconds to 5 minutes) and expert-level reasoning capabilities. Its core innovations include: the AF-CLAP audio encoder, trained with an improved contrastive loss on over 8 million audio–text pairs; the AudioSkills dataset, which consists of 4.2 million question–answer pairs covering seven categories of reasoning skills; and a three-stage curriculum training strategy including pretraining, fine-tuning, and long-audio fine-tuning.

**Audio-Flamingo-3 (AF3)** (Goel et al., 2025). Jointly developed by NVIDIA and the University of Maryland, this is a leading fully open-source LALM. Its main innovations include: the AF-Whisper unified audio encoder, which enables joint representation learning of speech, environmental sounds, and music; support for on-demand reasoning (e.g., chain-of-thought reasoning), multi-turn multi-audio dialogue, long audio understanding up to 10 minutes, and speech-to-speech interaction.

**Baichuan-Omni-1.5** (Li et al., 2025b). Developed by Baichuan Inc., this is a full-modality model capable of understanding text, image, audio, and video, as well as supporting end-to-end audio generation. Its main strengths include: a data processing pipeline that constructs and cleans approximately 500B high-quality multimodal data; the Baichuan-Audio-Tokenizer, designed to capture both semantic and acoustic features (implemented with an 8-layer RVQ structure and a 12.5 Hz frame rate); and a multi-stage training strategy consisting of image–text pretraining, image–audio–text joint pretraining, full-modality pretraining, and multimodal supervised fine-tuning.

**Gemma-3n** (Team, 2025). The Gemma 3n models are optimized for efficient execution on low-resource devices. They support multimodal input (text, image, video, audio) and generate high-quality text outputs. The series provides open weights for both pre-trained and instruction-tuned variants, and covers more than 140 natural languages. The Gemma 3n models employ selective parameter activation technology, which reduces resource requirements and allows the models to operate effectively at sizes of 2B or 4B parameters, although the total number of parameters is larger.

**Aero-1-Audio** (Li et al., 2025a). Aero-1-Audio is a compact audio model developed by LMMs-Lab as part of the Aero-1 series, the first generation of lightweight multimodal systems. Built upon the Qwen-2.5-1.5B language model, it achieves strong performance across speech recognition, audio understanding, and instruction-following benchmarks while remaining parameter-efficient. Trained within one day on 16 H100 GPUs with 50k hours of curated data, Aero demonstrates that efficient training is possible with high-quality samples. It further supports continuous audio inputs up to 15 minutes, a challenging setting for most existing audio models.

**GPT-4o** (Hurst et al., 2024). GPT-4o, released by OpenAI in August 2024, is an autoregressive universal model supporting arbitrary combinations of text, audio, image, and video as inputs, and

text, audio, and image as outputs. All modalities are processed by a single end-to-end trained neural network, enabling seamless multimodal integration and efficient inference across diverse tasks.

**Gemini-2.0-Flash-Lite** (Comanici et al., 2025a). Gemini-2.0-Flash-Lite, introduced by Google in April 2025, is the most cost-efficient member of the Gemini 2.0 family. It adopts a sparse Mixture-of-Experts Transformer architecture and leverages Trillium TPUs for training and inference. The model supports text, image, audio, and video inputs with a context window of 1,048,576 tokens, and produces text outputs of up to 8,192 tokens. Its design emphasizes scalability and latency efficiency for high-volume multimodal applications.

**Gemini-2.0-Flash** (Comanici et al., 2025a). Gemini-2.0-Flash is a natively multimodal model designed to power next-generation agentic systems. Compared with Gemini 1.5 Flash, it offers higher quality while maintaining comparable inference speed. It accepts text, image, audio, and video inputs with a 1,048,576-token context window and outputs text up to 8,192 tokens, with experimental image generation capabilities. Its architecture refines the sparse Mixture-of-Experts Transformer design with improved stability and optimization efficiency.

**Gemini-2.5-Flash** (Comanici et al., 2025a). Gemini-2.5-Flash is Google's first hybrid reasoning model, allowing developers to toggle reasoning on or off and allocate reasoning budgets for a trade-off between quality, cost, and latency. It supports text, image, audio, and video inputs with a 1M-token context window and generates text outputs up to 64K tokens. Based on a sparse Mixture-of-Experts Transformer with native multimodal support, it significantly outperforms Gemini-1.5-Pro on reasoning and multimodal benchmarks.

**Gemini-2.5-Flash-Lite** (Comanici et al., 2025a). Gemini-2.5-Flash-Lite extends the hybrid reasoning family with a cost-efficient design optimized for latency-sensitive tasks such as translation and classification. It provides improvements over Gemini-2.0-Flash-Lite in coding, mathematics, science, and reasoning, while supporting text, image, audio, and video inputs with a 1M-token context window and generating text outputs up to 64K tokens. Its sparse Mixture-of-Experts Transformer architecture balances efficiency with strong performance in large-scale multi-modal applications.

## F PROMPT

Here we present the prompt templates used for various tasks in our AUDIOMARATHON.

---

**Task: DESED sound event detection**

**System Prompt**
You are a helpful assistant that analyzes audio to detect and classify sound events.
Please listen carefully and select the most appropriate answer from the given choices.

**User Prompt Template**
`<audio>` Please listen to the audio and select the correct answer. **Reply with only the letter (A, B, C, or D)**. `{question}`
`A: {content of choice a}`
`B: {content of choice b}`
`C: {content of choice c}`
`D: {content of choice d}`

---

Figure 14: Prompt template for the SED task.

**Task: GTZAN music genre classification**

**System Prompt**

You are a helpful assistant that analyzes music audio to identify genres. Please listen to the audio carefully and classify the music genre.

**User Prompt Template**

`<audio>` Listen to this audio segment and identify the music genre based on what you hear.
**A: {content of choice a}**
**B: {content of choice b}**
**C: {content of choice c}**
**D: {content of choice d}**
**Respond with only the letter of your answer (A, B, C, or D).**

Figure 15: Prompt template for the MC task.

**Task: HAD audio authenticity detection**

**System Prompt**

You are a helpful assistant that analyzes audio to detect authenticity. Please listen to the audio carefully and determine if it is real or contains synthetic/fake content.

**User Prompt Template**

`<audio>` **{question}**
**A: {content of choice a}**
**B: {content of choice b}**
**Respond with only the letter of your answer (A or B).**

Figure 16: Prompt template for the SD task.

**Task: LibriSpeech ASR**

**System Prompt**

You are a helpful assistant that transcribes speech audio. Please listen carefully and provide the exact transcription of what is spoken in the audio.

**User Prompt Template**

`<audio>` Transcribe this audio accurately. **Remove hesitation words like 'um', 'uh'.**
**Your response should be formatted as follows: Spoken Content:**

Figure 17: Prompt template for the ASR task.

**Task: RACE reading comprehension**

**System Prompt**

Listen to this audio of a passage being read aloud, then answer the multiple-choice question based solely on the information from the audio.

**User Prompt Template**

`<audio>` Question: `{question}`

Options:
`A: {content of option A}`
`B: {content of option B}`
`C: {content of option C}`
`D: {content of option D}`

**Respond with only the letter of the correct option (A, B, C, or D).**

Figure 18: Prompt template for the SCR task.

**Task: SLUE named entity recognition**

**System Prompt**

You are a helpful assistant that analyzes audio to answer questions about named entities. Please listen to the audio and select the correct answer. **Reply with only the letter (A, B, C, or D)**.

**User Prompt Template**

`<audio> {question}`
`A: {content of choice a}`
`B: {content of choice b}`
`C: {content of choice c}`
`D: {content of choice d}`

Figure 19: Prompt template for the SER task.

**Task: TAU Urban Acoustic Scene Classification**

**System Prompt**

You are a helpful assistant that analyzes urban soundscape audio to identify acoustic scenes. Please listen to the audio carefully and classify the scene type.

**User Prompt Template**

`<audio>` Listen to this audio and identify the acoustic scene. Choose the most appropriate option.
**A: {content of choice a}**
**B: {content of choice b}**
**C: {content of choice c}**
**D: {content of choice d}**
**Respond with only the letter of your answer (A, B, C, or D).**

Figure 20: Prompt template for the ASC task.

**Task: VoxCeleb speaker gender classification**

**System Prompt**

You are a helpful assistant that analyzes audio to identify speaker characteristics. Please Listen to this audio and identify the speaker's gender.

**User Prompt Template**

`<audio>` Is this a male or female voice? **If it is a male, answer 'a'. If it is a female, answer 'b'. Answer with only 'a' or 'b'**

Figure 21: Prompt template for the SGR task.

**Task: VESUS emotion recognition**

**System Prompt**

You are a helpful assistant that analyzes audio to answer questions about emotions. Please listen to the audio carefully and select the correct answer.

**User Prompt Template**

`<audio> {question}`

**A) {content of choice a}**
**B) {content of choice b}**
**C) {content of choice c}**
**D) {content of choice d}**

**Please select the correct answer (A, B, C, or D).**

Figure 22: Prompt template for the ER task.

**Task: VoxCeleb speaker age classification**

**System Prompt**

You are a helpful assistant that analyzes speaker demographics. Please listen to this audio and identify the speaker's age group. Choose the most appropriate option: (a) Young Adult (18-30), (b) Early Career (31-40), (c) Mid Career (41-50), (d) Senior (51-70), (e) Elderly (71+). **Answer with only the letter (a, b, c, d, or e)**.

**User Prompt Template**

```
<audio> {question}
```

```
A) {content of choice a}
B) {content of choice b}
C) {content of choice c}
D) {content of choice d}
E) {content of choice e}
```

**Please select the correct answer (A, B, C, D, or E).**

Figure 23: Prompt template for the SAR task.

## G  LIMITATIONS AND FUTURE WORK

### G.1  LIMITATIONS

Our benchmark reflects practical choices in data sources and task design. Some datasets have license limits that restrict redistribution. The benchmark focuses on English and may not reflect cross-language behavior. We rely on automatic pipelines for audio concatenation and option generation, which can introduce bias if the source data has bias. While we test multiple long audio tasks, some domains and tasks are still underrepresented. Our evaluation covers common metrics but does not fully capture human preference or safety risks. Finally, we focus on inference efficiency methods at test time and do not include training time efficiency or energy use.

### G.2  FUTURE WORK

We plan to run systematic hyperparameter searches at key encoder and decoder layers to measure sensitivity and find settings that preserve temporal detail while improving efficiency. We will evaluate more compression and acceleration methods, including stronger token selection methods and better cache policies, and test transfer from text and vision methods to audio. We will add more tasks and languages, broaden source datasets, and release tools for reproducible data building and evaluation. We also plan to study human evaluation for long audio tasks and extend metrics that measure temporal continuity and memory. Finally, we will report energy and cost to give a fuller view of efficiency.

## H  USE OF LLMs

In this study, we utilized large language models (LLMs) to perform grammar checking and to polish certain sentences for improved clarity and fluency, without altering the original meaning of the text. Auxiliary AI coding tools are used for debugging and analyzing code errors, as well as assisting in code implementation, with the main code being constructed and carefully reviewed by humans.

