# OpenReview forum: "AudioMarathon: A Comprehensive Benchmark for Long-Context Audio Understanding and Efficiency in Audio LLMs"
_ICLR.cc/2026/Conference — ICLR 2026 Conference Withdrawn Submission_

### Official Review · Reviewer_Ct46 · 2025-10-26

**Soundness:** 3
**Presentation:** 3
**Contribution:** 2
**Rating:** 4
**Confidence:** 4

**Summary:**

This paper introduces AudioMarathon, a comprehensive benchmark for long audio understanding, spanning 10 audio tasks. The authors evaluate several sota LALMs on this benchmark, revealing that the methods struggle more as audio lengths increases.  They also study various acceleration techniques, such as token pruning and KC-cache eviction, to analyze their tradeoffs.

**Strengths:**

- The paper is well-written and the objectives are clearly stated.
- The paper includes several results.
- The focus of the paper in evaluating LALMs on long-duration clips is of sufficient importance.

**Weaknesses:**

- The papers claims that most of the existing benchmarks entail short audio clips, and therefore they don't capture "complexity of real-world scenarios such as meetings, podcasts, and extended dialogues" (lines 107-108). However, the duration of the clips in AudioMarathon is in the range [90,300]seconds, which in my opinion is quite shorter than the average duration of meetings/podcasts. This is why the benchmark provided in "Orevaoghene Ahia et al., BLAB: Brutally Long Audio Bench, 2025" seems more appropriate as the average audio duration is 51minutes.
- The results about compression and peak memory usage are somewhat expectable and they do not add much value to the paper.
- The paper lacks multiple details and clarifications --> see questions below.
-  It's not clear to me how the proposed "Frame" compression method works in practice. The description at lines 268-268 is insufficient.
- In Figure 7, it seems like the SnapKV line is never displayed in the 4 plots, and so does it seem for KNorm method on figures (b), (c), and (d). Are they completely overlapping with the other methods or what?

**Questions:**

**Questions**:
- Regarding step 2/3, can the authors elaborate more on the logic behind the concatenation of and merging of clips to have longer sequences and the corresponding “tool design”? The description in Section 2.2 is a bit vague.
- Regarding step 5, why did the authors only review 10% of the data? How many samples needed to be fixed? Unless the failure ratio is extremely low, then I think it would be better to double check all the samples.
- Why Flamingo-2 achieves only 1% of WAR? It seems a big outlier.
- Why in Table 4, random pruning for Qwen2.5-Omni-3B oftentimes perform better than the other methods which perform more accurate compressions?
- In the paragraph “How do token pruning methods affect performance in LALMs?”, how does the authors explain the fact that Qwen2.5-Omni-3B “benefits from token pruning”? Do they mean that the model benefits when the ratio is 30% or even up to 90%? It seems to deteriorate rather than improve, and this makes sense given the high compression ratio.
- Can the authors provide some examples of failure cases in some of the considered tasks? Or at least compare when some models fail/succeed. Based on how and where the models make the mistakes, I think it would be easier to devise better techniques to improve their capabilities on long-context scenarios.


**Typos**:

- Duplicated reference: Qwen2.5-omni technical report by Jin Xu et al., 2025.
- Missing full stops on all image captions.
- Missing acronym definition: MCQ.
- In Table 3: Close-source —> Closed-source.

---

> ### Author Response · Authors · 2025-11-21
> **The Preliminary Rebuttal (1/2)**
>
> Dear Reviewer Ct46,
>
> We apologize for any confusion caused by our presentation.  We will explain them to you sequentially.
>
> >**W1. Length of average audio duration.**
>
> We appreciate your thoughtful comparison with BLAB [1]. Our remark about “short audio clips” is made relative to the context limits of current LALMs: in practice, most can handle at most 30 minutes of mixed audio–text input, so we deliberately target 90–300s as a “difficult but still runnable” regime where many models can be evaluated fairly. We fully agree that real meetings and podcasts are longer, and we have implemented and plan to release an additional 30-minute meeting audio summarization subset as a stress-test extension. We also note that BLAB’s hour-level sequences are so demanding that long-audio results there have only been reported on Gemini-2.0-Flash and Gemini-2.0-Pro [1], whereas AudioMarathon is designed to remain broadly runnable under today’s context and memory budgets.
>
> >**W2. The results about compression and peak memory usage are somewhat expectable.**
>
> We totally understand that these trends may look intuitive. We simplified the expression and length of this section into a small part.
>
> >**W3. Explanation on FRAME method.**
>
> Section 4.2 now provides a more detailed and precise exposition of the FRAME algorithm:
>
> FRAME is a training-free, time-aligned pruning method that operates only on the audio encoder tokens while leaving non-audio context untouched.
> Given a mixed-modality token sequence, we denote the contiguous audio span by **$[t_0, t_0 + L)$** and a pruning ratio **$r \in [0, 1)$**.
>
> FRAME first computes the target budget of retained audio tokens $K$ using the formula:
>
> $$K = \max\{1, \lfloor L (1 - r) \rfloor\}$$
>
> where:
>
>   * $L$ is the length of the audio span.
>   * $r$ is the pruning ratio.
>   * $K$ is the number of audio tokens to be retained.
>
> FRAME then performs uniform sampling within the audio span by selecting indices at equal spacing, and finally aggregates these sampled audio indices with the untouched prefix and suffix tokens into a single keep set used to gather hidden states and recompute the causal mask before the decoder layer.
>
> >**W4. Figure 7 details.**
>
> As you observed, most strategies overlap almost completely in Figure 7. We have optimized the expression in the revised manuscript to state this more clearly.
>
> >**Q1. Explanation on tool design.**
>
> Our tool design is task-specific rather than one-size-fits-all, and we provide the detailed implementations in the Supplementary Material. For example, for Acoustic Scene Classification on TAU Urban Acoustic Scenes, we concatenate raw clips into longer sequences under custom duration constraints and then use a small script to infer scene labels from directory and filename patterns, form four-way MCQs by sampling distractor scenes from a canonical option list, and write the audio path, label, and ID into a JSON file. Similar task-specific tools are used for other subsets.
>
> >**Q2. Data Review.**
>
> We first estimated the failure rate of the automated pipeline in a small-scale trial and found the error proportion to be extremely low. Based on this, we manually reviewed a random 10% of the full sample set and recorded the number of samples requiring correction. Given this estimate, even a full manual review of the remaining 90% would have a very limited impact on the overall statistical conclusions but would incur very high costs. Therefore, we chose this trade-off between quality and cost. We reviewed the reliability of the remaining data and found that only a portion of the ER (Emotion Recognition) tasks present the issue that "It is hard even for humans to distinguish if the speaker is happy." We will add a CSV file to the new version of the dataset to declare this issue and suggest that users decide whether to use the controversial data for specific speakers.
>
> >**Q3. Why Flamingo-2 achieves only 1% of WAR? It seems a big outlier.**
>
> Flamingo‑2 is designed for long‑context audio understanding, but this version does not include an ASR component [2]. Such capability is only added in Flamingo‑3 [3]. So we included it to demonstrate that without targeted ASR alignment, LALMs suffer severe degradation in long-form transcription tasks.

---

> ### Author Response · Authors · 2025-11-21
> **The Preliminary Rebuttal (2/2)**
>
> >**Q4. Explanation of why random pruning often performs better**
>
> We apologize for the lack of clarity in our original discussion of this section. We have incorporated a detailed discussion on this issue in the revised version. In brief:
> The better performance of random pruning for Qwen2.5-Omni-3B stems from two primary factors rooted in the model's encoding characteristics and the limitations of current advanced pruning methods:
>
> * **Redundancy as Noise:** Qwen2.5-Omni-3B's audio tokenization contains substantial uninformative redundancy, potentially even noise. Consequently, random pruning functions analogously to a **regularization or denoising mechanism**, filtering out non-essential tokens and allowing the model to concentrate on core semantic information.
> * **Preservation of Temporal Granularity:** Sophisticated methods transferred from the vision modality (e.g., DART, FastV) rely on redundancy or attention metrics that inadvertently discard **rare temporal segments** critical for fine-grained tasks like ASR.
>
> Ultimately, this phenomenon highlights the high degree of redundancy in current LALMs and indicates that directly migrating pruning methodologies from vision or text domains is insufficient for the distinct temporal requirements of audio.
>
> >**Q5. Explanation of benefits from token pruning.**
>
> When the ratio is 30%, we observe significant inference acceleration and memory savings across multiple tasks. Meanwhile, the performance of Qwen2.5-Omni-3B remains stable. This demonstrates that at moderate pruning ratios, reasonably designed pruning strategies can achieve a superior trade-off between efficiency and performance.
>
> >**Q6. More examples of failure cases.**
>
> We have already provided them in Appendix B. We understand your concern that the current examples may be insufficient. In the revised version of Appendix B, we have added more representative failure cases and provided corresponding analyses and explanations.
>
>
> [1] BLAB: A Benchmark for Long Audio Understanding. arXiv 2505.
>
> [2] Audio Flamingo 2: An Audio-Language Model with Long-Audio Understanding and Expert Reasoning Abilities. ICML 2025.
>
> [3] Audio Flamingo 3: Advancing Audio Intelligence with Fully Open Large Audio Language Models. arXiv 2507.

---

> ### Comment · Reviewer_Ct46 · 2025-11-24
> **Follow-up questions**
>
> I thank the authors for addressing my questions and reported weaknesses. I have a few additional comments and questions.
>
> **1)** Does the current PDF paper reflects the changes you have applied during the rebuttal? For example, you wrote that *"Section 4.2 now provides a more detailed and precise exposition of the FRAME algorithm:"*. However, I do not see section 4.2 in the current version, and obviously the formulation of the proposed FRAME algorithm. Can you highlight in blue or red the modified sections/sentences in the updated PDF version so we can compare with the original version? Thank you.
>
> **2)** Still regarding the FRAME algorithm, it is not yet clear to me the real motivation behind its introduction in the paper. This is buttressed by the fact that the authors **do not mention** it neither in the abstract nor in the full introduction. It is an additional token strategy used to compare various LLMs, but its purpose is quite limited in my opinion. Furthermore, the authors claim that *"Frame consistently outperforms other methods across different latency constraints."* (caption of Figure 5). So it seems like FRAME is part of the novelty and contribution of the paper itself, which is not reported in the abstract and introduction.
>
> **3)** The authors have studied efficiency strategies to mitigate the issue of dealing with many tokens. However, I do not see any comments regarding **how** current LALMs could be equipped with better long-form audio capabilities. All current models are pretty far from obtaining results comparable with human evaluations, so which techniques or strategies could be harnessed to improve their long-form audio capabilities? Is there any recent method which tries to accomplish this? I believe such a discussion could be worth being included in the paper to give the reader some possible future directions on how to improve the current LALMs.

---

> > ### Author Response · Authors · 2025-11-26
> > **Response of follow-up questions**
> >
> > Dear Reviewer Ct46,
> >
> > Thank you once again for reviewing our paper and providing valuable feedback. We have carefully considered your suggestions and made multiple revisions to enhance the clarity, depth, and contribution of the paper.
> > >Q1+Q2. New Version and our revision of pdf.
> >
> > We apologize that the revised PDF was not uploaded previously; it is now available.
> > Major revisions in the document have been highlighted in blue. We have also refined the overall phrasing and optimized previously overly general expressions for enhanced clarity and precision.
> >
> > For instance, the caption for Figure 5 has been updated to the more precise and detailed description: "Comparisons of latency and performance trade-off for the Qwen2.5-Omni-3B model under different token pruning strategies across four representative datasets. Frame outperforms other methods on speech content extraction tasks across different latency constraints in almost all cases."
> >
> > >Q2 The proposition and motivation of FRAME
> >
> > First, we would like to present performance data across various pruning ratios for tasks categorized under speech content extraction, specifically highlighting the competitive advantages of the FRAME method over conventional baselines.
> > | Method | Prune Ratio | ASR | SCR | SER |
> > |:---|:---|:---:|:---:|:---:|
> > | **Vanilla** | 0 | 97.2 | 83.2 | 25.2 |
> > | DART | 10% | 87.4 | 80.5 | 23.3 |
> > | Random | 10% | 92.5 | 81.8 | 26.7 |
> > | Frame | 10% |**93.9**| **82.0** | **27.7** |
> > | DART | 30% | 81.4 | 74.2 | 23.2 |
> > | Random | 30% | 88.4 | 80.3 | 26.5 |
> > | Frame | 30% | **92.2** | **80.9** | **26.8** |
> > | DART | 40% | 73.2 | 70.7 | 24.0 |
> > | Random | 40% | 83.0 | 80.2 | 26.8 |
> > | Frame | 40% | **90.5** | **81.4** | **27.2** |
> > | DART | 50% | 73.3 | 67.1 | 24.3 |
> > | Random | 50% | 74.1 | 78.1 | **25.3** |
> > | Frame | 50% | **90.1** | **81.1** | 24.8 |
> > | DART | 60% | 62.8 | 64.6 | 23.1 |
> > | Random | 60% | 59.7 | **75.3** | 24.2 |
> > | Frame | 60% | **82.2** | **75.3** | **25.6** |
> >
> > As demonstrated, while FRAME achieves strong performance, often surpassing baselines, its primary contribution is not as a best-performing comparative model for Large Audio Language Models (LALMs). Instead, FRAME is an analytical framework that highlights two critical insights for audio token pruning:
> > * **Task-Aware Pruning**: Effective audio token pruning necessitates task-aware design rather than a universal strategy.
> >
> > * **Audio Modality Specificity**: Directly migrating existing MLLM token pruning methods, which are inherently biased towards visual and textual data, is suboptimal for audio. Novel methodologies, derived from inherent aud
> >
> > In this regard, FRAME serves as an **exploratory paradigm to guide future development** towards more comprehensive and audio-centric token pruning methods, explaining its positioning as a foundational discussion rather than a key performance metric in the Abstract.
> >
> > >Q3 How current LALMs could be equipped with better long-form audio capabilities
> >
> > We would like to thank for your constructive suggestion regarding our future work direction.
> > We thank the reviewer for this suggestion. Below is three latest methods aiming at improving long-form audio capabilites.
> >
> > * **Improved encoding granularity of LALMs**
> > Recent leading LALMs have already started to explicitly address the redundancy of audio token embeddings at the architectural level. For example, in the audio encoding design of Qwen3-Omni, the dedicated audio encoder increases the temporal span represented by each token from 40 ms in Qwen2.5-Omni to 80 ms [1, 2], effectively halving the token rate.
> >
> > * **Chunk-wise modeling with adaptive token compression.**
> > Process audio in short local chunks then aggregate into coarser segment representations. Insert token-merging after the local encoder so the heavy cross-attention operates on a compressed, semantically-rich sequence, preserving temporal detail while reducing quadratic cost [3].
> >
> > * **Memory-augmented context reuse.**
> > Maintain compact episodic summaries that the decoder can read. This avoids re-encoding distant audio and preserves long-range context with fixed compute and bounded memory[4] .
> >
> >
> > We will add this part into our disscussion as part of our future work.
> >
> > [1] Qwen2.5-Omni Technical Report. arXiv 2503.
> >
> > [2] Qwen3-Omni Technical Report. arXiv 2509.
> >
> > [3] ChunkFormer: Masked Chunking Conformer For Long-Form Speech Transcription.  ICASSP 2025.
> >
> > [4] WavRAG: Audio-Integrated Retrieval Augmented Generation for Spoken Dialogue Models. ACL 2025.

---

> > > ### Comment · Reviewer_Ct46 · 2025-11-26
> > >
> > > Thank you for the additional clarifications. I have no further questions. While most of my concerns and questions have been clarified, I'll take some more time to analyze in more detail the contributions of the paper and the impact that such a dataset can have on the community. I'll be happy to discuss about this with the other reviewers and ACs as well.
> > >
> > > Kind regards,
> > >
> > > Reviewer Ct46

---

> > > > ### Author Response · Authors · 2025-11-27
> > > > **Reply for Reviewer Ct46**
> > > >
> > > > Dear Reviewer Ct46,
> > > >
> > > > We are deeply grateful for your prompt response and your continuous engagement throughout this discussion period. It is truly encouraging to hear that our previous responses have successfully clarified most of your concerns.
> > > >
> > > > We sincerely appreciate your willingness to dedicate further time to evaluate the contributions and potential community impact of AudioMarathon.
> > > >
> > > > We remain fully available should any new questions arise during your analysis.
> > > >
> > > > We genuinely hope that, upon further reflection on the value this work brings to the multimodal research community, maybe you could consider raising your score to support the acceptance of our paper.
> > > >
> > > >
> > > > Best regards,
> > > >
> > > >
> > > > The Authors

---

### Official Review · Reviewer_tjz8 · 2025-10-27

**Soundness:** 2
**Presentation:** 2
**Contribution:** 2
**Rating:** 4
**Confidence:** 4

**Summary:**

This paper introduces AUDIOMARATHON, a long-audio benchmark that aggregates 10 tasks spanning speech, environmental audio, and music domains. By studying the performance of open-source and closed-source models on AUDIOMARATHON, the paper shows that current LALMs are still underperforming with long input. Moreover, this paper studies inference-time acceleration techniques for LALMs, including token pruning and KV-cache eviction, and shows their trade-offs to performance.

**Strengths:**

1. Focus on long-form audio and well-thought-out tasks spanning across each domain in audio.
2. Comprehensive experiment results comparing open-sourced and closed-sourced LALMs
3. Analysis on performance trade-off focusing on inference-time efficiency
4. Clear presentation in the first half of the paper for the dataset specification

**Weaknesses:**

Formulation of the “long-audio” datasets needs clearer specification.
1. The work appears to add limited new “long-audio” material. Figure 2 suggests that all tasks are long-form and created by concatenation, which is misleading. Appendix D indicates that the only newly introduced long-form set is RACE, and its audio is produced by TTS from a text dataset. For environmental audio and music tasks, it seems the audio is taken directly from the original sources.
2. For the “multi-hop” or “complex reasoning” component, namely RACE, the evaluation largely probes ASR capability and downstream text understanding. An audio-specific, more acoustic-centered multi-hop design would better substantiate the “complex reasoning” claim.

Presentation of experiments and the path from results to conclusions is confusing. The paper would offer more actionable guidance for LALM development with tighter exposition, more focused ablations, and conclusions that are directly supported by the evidence.
1. Frame pruning is under-specified. The text motivates time-aligned pruning and claims that Frame preserves rare short acoustic events, but it does not provide a precise algorithmic description that enables reproduction, for example window size, stride, scoring rule, tie breaking, and any layer-wise schedules. Appendix A mainly explains why pruning occurs at the second layer rather than detailing how Frame operates.
2. KV-cache eviction analysis is limited. Figure 7 shows peak memory across policies, yet the curves look similar and appear driven primarily by the number of tokens. The paper does not report accuracy trade-offs under different eviction settings.
3. Length attribution is missing. The claim that “current models fail at long audios” would be stronger with a length-controlled comparison. The paper aggregates minute-scale results and human baselines, but it does not include short-context baselines for the same tasks or a duration sweep. Without a length-matched ablation, it is difficult to attribute errors to long context rather than task or domain effects.
4. Figure-level clarity issues. The caption of Figure 9 does not make clear which pruning method each panel uses, and the nearby text only mentions “varying token pruning ratios” and “maximum F1 across settings,” which prevents mapping curves to methods. In Figure 5, the caption states that “Frame consistently outperforms other methods,” which seems overstated, since in later segments of ASC and SGR other methods match or surpass Frame at higher token counts.
5. Evidence for “task-aware audio token compression is essential” is insufficient. The Results section makes a broad necessity claim, but the curves mainly show about a 20 percent latency reduction in several regimes. The paper does not provide controlled accuracy-at-fixed-latency comparisons against strong baselines, nor latency-at-fixed-accuracy tables that would justify essentiality rather than usefulness.
6. Actionable request tied to the claim (line 365). Where the paper asserts that token compression is “essential,” please add controlled accuracy-at-fixed-latency or latency-at-fixed-accuracy tables. The current evidence supports that certain pruning methods can maintain similar accuracy with about a 20 percent latency drop. It does not yet support the stronger claim that token compression is essential.

**Questions:**

See Weaknesses.

---

> ### Author Response · Authors · 2025-11-21
> **The Preliminary Rebuttal (1/2)**
>
> Dear Reviewer tjz8,
>
> We apologize for any lack of clarity for our presentation.
> >**W1. Formulation of the “long-audio” datasets needs clearer specification.**
>
> We understand your concern about the construction process of long-form audio. We would like to clarify that none of the audio is taken directly from the original sources. We use indirect information, such as filenames and JSON annotations, to align the audio in the subsets of various datasets, and build long-form audio based on these alignments. We build individual scripts to achieve this goal.
>
> >**W2. Multi-hop reasoning emphasis.**
>
> For the “multi-hop” or “complex reasoning” component, namely RACE, the evaluation largely probes text understanding capability. We agree that an audio-specific, more acoustic-centered multi-hop design would better substantiate the “complex reasoning” claim. In this work, we view RACE as a first step connecting long-form audio to text understanding, and we are actively conceptualizing how to structure a more refined acoustic multi-hop reasoning design.
>
> >**Q1. FRAME specification & reproducibility.**
>
> We apologize that the description of FRAME was hard to follow and not clear enough in the original submission.
> Section 4.2 now provides a more detailed and precise exposition of the FRAME algorithm:
> FRAME is a training-free, time-aligned pruning method that operates only on the audio encoder tokens while leaving non-audio context untouched.
> Given a mixed-modality token sequence, we denote the contiguous audio span by **$[t_0, t_0 + L)$** and a pruning ratio **$r \in [0, 1)$**.
>
> FRAME first computes the target budget of retained audio tokens $K$ using the formula:
>
> $$K = \max\{1, \lfloor L (1 - r) \rfloor\}$$
>
> where:
>
>   * $L$ is the length of the audio span.
>   * $r$ is the pruning ratio.
>   * $K$ is the number of audio tokens to be retained.
>
>
> >**Q2. KV-cache eviction analysis.**
>
> Thank you for raising this point. In our experiments, the majority of cache eviction during prefilling has negligible impact on QA-style tasks in AudioMarathon, as **the performance changes are within 1%** across policies and compression ratios on Qwen2,5-Omni-3B, as shown below:
> | **Method** | **Ratio** | **SER** | **SCR** | **ASR** | **SED** | **MC** | **ASC** | **SD** | **ER** | **SAR** | **SGR** |
> |:----------:|:---------:|:-------:|:-------:|:-------:|:-------:|:------:|:-------:|:------:|:------:|:-------:|:-------:|
> | Vanilla | 0.0 | 25.2 | 82.3 | 94.7 | 70.2 | 97.4 | 69.3 | 67.3 | 39.6 | 29.1 | 97.2 |
> | KNORM | 0.3 | 25.2 | 82.3 | 91.2 | 70.2 | 97.4 | 69.3 | 67.3 | 39.6 | 29.1 | 97.2 |
> | KNORM | 0.5 | 25.2 | 82.3 | 35.7 | 70.2 | 97.4 | 69.3 | 67.3 | 39.6 | 29.1 | 97.2 |
> | KNORM | 0.9 | **24.9** | **82.0** | **0.0** | **70.0** | **96.8** | **68.9** | **66.8** | **38.3** | **28.6** | **97.0** |
> | RANDOM | 0.3 | 25.2 | 82.3 | 77.0 | 70.2 | 97.4 | 69.3 | 67.3 | 39.6 | 29.1 | 97.2 |
> | RANDOM | 0.5 | 25.2 | 82.3 | 21.4 | 70.2 | 97.4 | 69.3 | 67.3 | 39.6 | 29.1 | 96.9 |
> | RANDOM | 0.9 | **25.4** | **82.1** | **0.0** | **69.6** | **97.9** | **69.0** | **66.3** | **38.9** | **27.5** | **96.4** |
> | SNAP | 0.3 | 25.2 | 82.3 | 32.8 | 70.2 | 97.4 | 69.3 | 67.3 | 39.6 | 29.1 | 97.2 |
> | SNAP | 0.5 | 25.0 | 82.3 | 0.0 | 70.2 | 97.4 | 69.3 | 67.3 | 39.6 | 29.1 | 97.2 |
> | SNAP | 0.9 | **25.0** | **81.8** | **0.0** | **69.9** | **96.8** | **68.6** | **67.0** | **39.0** | **26.5** | **97.2** |
> | TOVA | 0.3 | 25.2 | 82.3 | 88.6 | 70.2 | 97.4 | 69.3 | 67.3 | 39.6 | 29.1 | 97.2 |
> | TOVA | 0.5 | 25.0 | 82.3 | 42.5 | 70.2 | 97.4 | 69.3 | 67.3 | 39.6 | 29.1 | 97.2 |
> | TOVA | 0.9 | **24.9** | **81.9** | **0.0** | **69.0** | **96.3** | **68.3** | **66.2** | **37.8** | **24.5** | **96.8** |
>
> So Figure 7 mainly reflects peak memory as a function of the number of cached tokens. These observations are consistent with prior work on text and multimodal QA [1, 2]. For completeness, we will explicitly summarize this near–no-loss behavior in the main text and refer readers to Appendix A.3 for the detailed theoretical discussion and additional results, including the contrasting degradation we observe on ASR-type tasks with long decoding horizons on Qwen2.5-Omni-3B.

---

> ### Author Response · Authors · 2025-11-21
> **The Preliminary Rebuttal (2/2)**
>
> >**Q3. Length attribution is missing.**
>
> We have constructed a short-audio version of the dataset and performed tests on the three main models evaluated: Qwen2.5-Omni-3B, Phi4-MM, and Aero-1-Audio. The results are shown as follows:
> | Model\Datasets | SER | SCR | ASR | SED | MC | ASC | SD | ER | SAR | SGR | Avg. |
> | :--- | :---: | :---: | :---: | :---: | :---: | :---: | :---: | :---: | :---: | :---: | :---: |
> | Qwen2.5 | 38.6 | 89.1 | 96.9 | 66.2 | 97.3 | 58.0 | 33.2 | 40.2 | 29.3 | 92.6 | **64.1** |
> | Phi4MM | 14.0 | 75.8 | 96.8 | 48.4 | 61.8 | 29.2 | 2.5 | 27.3 | 23.8 | 91.1 | **47.1** |
> | Aero-1 | 13.8 | 63.2 | 97.6 | 51.1 | 73.8 | 42.8 | 42.7 | 12.5 | 11.1 | 33.6 | **44.2** |
>
> This result demonstrates:
> * **Distraction in Entity Tracking**: In the short-audio setting, Qwen2.5-Omni-3B exhibited a significant improvement (+13.4) in Speech Entity Recognition (SER), and (+6.8) in Speech Content Reasoning (SCR). This indicates that redundant information inherent in long audio sequences tends to dilute critical cues when attempting to localize specific speakers or details. Conversely, short-audio environments significantly mitigates the complexity of reasoning.
> * **Stratification in Robustness**: For the lightweight model Aero-1-Audio (1.5B), the short-audio environment facilitated a dramatic recovery in ASR performance, bounding from a catastrophic failure in the long-audio setting (43.7) to 97.6. This indicates severe bottlenecks in positional encoding or memory retention within its long-context encoding mechanism. In contrast, Qwen2.5-Omni-3B maintained high ASR performance across both long and short contexts (94.7 vs. 96.9), exhibiting superior robustness in processing long sequences.
>
> >**Q4-Q6. Figure clarity issue.**
>
> The caption of Figure 9 shows the result of the maximum F1-score achieved across tested pruning settings. We want to clarify that we mentioned this in our paper. We understand that you feel the annotation was not clear enough. We have revised it and bolded this point. We would like to clarify that Table 4 in our manuscript effectively serves as an "Accuracy-at-Fixed-Latency" comparison. We have added lines on the x-axis for readers to compare performance under same latency as your suggestion.
>
> >**Q5-Q6. Evidence for “task-aware audio token compression**
>
> We would like to thank for your thoughtful and thorough comments. To clarify, we add a **Task-Aware Hybrid Score** which is the maximum  of FRAME, DART, and FastV:
>
> | Pruning Ratio | Random Baseline (Avg) | Task-Aware Hybrid Score | Performance Gain (Delta)
> | :--- | :--- | :--- | :--- |
> | **Light (↓ 30%)** | 66.8 | **67.5** | **+0.7** |
> | **Medium (↓ 60%)** | 61.7 | **65.6** | **+3.9** |
> | **Extreme (↓ 90%)** | 50.4 | **58.3** | **+7.9** |
> ---
>
>
> A comprehensive analysis reveals that the necessity of a task-aware strategy is significantly and positively correlated with the compression ratio. In the light pruning regime (30%), the algorithmic hybrid marginally outperforms the random baseline (+0.7 points) by leveraging the inherent redundancy of audio, suggesting that complex selection mechanisms are superfluous at this stage. However, under medium pruning (60%), algorithmic advantages become established (+3.9 points), driven primarily by the Frame strategy’s effective preservation of critical information in temporally sensitive tasks. Crucially, under extreme pruning (90%), the hybrid strategy maximizes gains (+7.9 points): DART successfully circumvents Frame’s catastrophic collapse in ASR (restoring the score from 0.0 to 62.9), while Frame maintains its lead in semantic understanding. This trend strongly indicates that as compression intensifies, single-strategy approaches falter.
>
> [1] Dynamic-LLaVA: Efficient Multimodal Large Language Models via Dynamic Vision-language Context Sparsification. ICLR 2025.
>
> [2] KV-Compress: Paged KV-Cache Compression with Variable Compression Rates per Attention Head. arXiv 2410.

---

> > ### Comment · Reviewer_tjz8 · 2025-11-26
> >
> > Thank you for the detailed rebuttal and for updating the draft. I appreciate the additional context, but I still find the concrete data creation procedure insufficiently specified, both for the long audio benchmark and for the new short audio experiment in Q3.
> >
> > In particular, could you please clarify the following, ideally in the revised appendix:
> > 1. Construction of long audio for each task.
> > For each of the ten tasks, please spell out the exact pipeline you use to construct the long audio examples, including at least
> >  - the original dataset and split you start from,
> >  - which metadata or annotations you use for alignment (for example filenames, JSON fields, timestamps),
> >  - how you transform or concatenate the underlying clips into a single long recording (ordering rule, segment length distribution, whether you insert silences or cross fades, resampling and channel handling, maximum total duration), and
> >  - how task labels are propagated or recomputed for the resulting long audio files.
> > A concise per-dataset description or a summary table would make the benchmark much more transparent and reproducible.
> > 2. Definition of the short audio condition in Q3.
> > For the new length attribution experiment, could you clarify exactly what data is used to obtain the short audio results in the table? In particular
> >  - how you derive the short audio version from the long audio benchmark for each task (do you revert to the original datasets, or sample segments from your long audio files),
> >  - the typical duration range for the short audio clips for each task,
> >  - whether the short and long conditions are matched example by example (same underlying content, different truncation) or drawn from different subsets.
> > At present I cannot tell what concrete data configuration underlies the numbers reported for Qwen2.5, Phi4 MM, and Aero 1 in Q3, which makes it hard to interpret how much of the gain is due purely to shorter context rather than a change of dataset or sampling protocol.

---

> ### Author Response · Authors · 2025-11-28
> **Construction pipeline and experiment results**
>
> Dear Reviewer tjz8,
>
> We thank the reviewer for the insightful comments regarding the reproducibility of our benchmark construction and the experimental design. We agree that transparency regarding the data pipeline is essential to interpreting the ablation results. We are going to update Appendix D to include a detailed specification of the construction process. Below, we provide the specific clarifications requested.
>
> * **Construction of Long Audio.**
> For all 10 tasks in AUDIOMARATHON, we employ a consistent "Concatenation-based" pipeline to construct minute-scale long-form audio from original short-clip datasets.
>
>
> * **Source & Split.** We strictly utilize the official test splits (or validation splits where test labels are withheld) of the source datasets to ensure no data leakage.
>
>
> * **Concatenation Strategy.** We utilize a custom script (Step 3 in our pipeline) to concatenate individual clips. For a given task, we randomly sample clips from the source pool and concatenate them sequentially until the target duration is reached. In fact, most audio files in original datasets are splited into short audios. For example, it is reasonable and clear if we mix "airport_London_001_1" with "airport_London_001_2".
>
>
> * **Duration.** As stated in the Abstract and Section 2.2, the resulting files for the main benchmark range from 90.0 to 300.0 seconds.
>
>
> * **Boundary Handling.** We do not insert artificial silence or cross-fades between segments. We intentionally preserve hard boundaries to strictly test the model's ability to handle abrupt context shifts and maintain temporal attention without resetting.
>
> * **Label Propagation.** For Classification/QA: The ground truth for the target segment is preserved, and its position is randomized within the long sequence to mitigate positional bias.
> For ASR, we concatenate the transcripts of the constituent clips in the temporal order of the audio.
>
> * **Specific Metadata & Alignment Sources.**
> We align audio with annotations using methods specific to the format of the original datasets:
>
>
> | Category | Datasets | Alignment & Processing Method |
> | :--- | :--- | :--- |
> | **File/Directory-based** | DESED, GTZAN, HAD | Event classes and genre labels are derived directly from the original filenames and directory structures. |
> | **Metadata Parsing** | SLUE, TAU, VESUS | Official JSON or CSV metadata files are parsed to map audio filenames to their specific timestamps and annotation labels. |
> | **Speaker-Centric Aggregation** | LibriSpeech, VoxCeleb |  **LibriSpeech**: Multiple short clips from the same speaker and chapter are concatenated to form continuous long-form audio files. **VoxCeleb**: Official identity CSV files are utilized to extract speaker age and gender labels. |
> | **Text-to-Speech Generation** | RACE | Audio is generated using the Kokoro-82M TTS system, with ground truth (QA pairs) derived directly from the original JSON fields. |
>
> >Q3 Definition of Short Audio Condition.
>
> Regarding the "Short Audio" ablation study presented in Table 8 (Appendix A.4), we wish to clarify the construction method to address the concern about data protocol differences:
>
> * **Consistent Protocol.** To ensure a fair comparison, the "Short Audio" condition was not simply the raw original files. We applied the exact same concatenation and processing pipeline used for the main benchmark.
>
> * **Duration Constraint.** The only variable changed was the target duration constraint. For the "Short Audio" condition, the concatenation script was set to limit the output length to the 15s to 30s interval.
>
> Therefore, the performance gaps reported are attributable specifically to the context length and the increased difficulty of temporal reasoning over minute-scale durations, rather than differences in sampling protocols or audio formatting.

---

### Official Review · Reviewer_Eib9 · 2025-10-29

**Soundness:** 4
**Presentation:** 3
**Contribution:** 4
**Rating:** 6
**Confidence:** 3

**Summary:**

The authors introduce AudioMarathon, a novel and comprehensive benchmark designed to evaluate Large Audio Language Models (LALMs) on long-form audio. The work is motivated by the significant gap in existing benchmarks, which primarily use short audio clips and thus fail to test models on realistic, minute-scale tasks. AudioMarathon features audio inputs from 90 to 300 seconds across 10 diverse tasks covering speech, music, and environmental sounds. Using this benchmark, the paper conducts a large-scale evaluation of 16 state-of-the-art LALMs, revealing significant performance degradation as audio length increases and a substantial gap compared to human performance, particularly on tasks requiring temporal reasoning. A key part of the study is a systematic analysis of inference efficiency techniques like token pruning and KV-cache eviction, highlighting the critical trade-offs between computational cost and model accuracy for long-context audio processing.

**Strengths:**

1. Significant Motivation: The motivation for AudioMarathon is highly significant. Most existing benchmarks and datasets for large audio models focus on short audio clips, whereas many real-world applications require the processing of long-form audio. This benchmark effectively addresses that gap.
2. Comprehensive and Diverse Scenarios: The benchmark includes 10 different sub-tasks that cover a wide range of audio-related scenarios, with tasks designed to evaluate both semantic and acoustic capabilities.
3. Robust Data Pipeline: The paper details a well-designed data pipeline that includes source selection, automated construction, and manual verification, which together ensure the reliability and quality of the benchmark.
4. Sufficient Model Evaluation: The study provides a thorough evaluation by testing 16 different audio models and including human evaluation scores as a baseline. This clearly demonstrates the current capabilities of models on this new benchmark.
5. Valuable Exploration of Efficiency Optimization: In the context of long audio, efficiency is critical. The paper's exploration of token pruning and KV-cache eviction demonstrates that while these techniques can reduce memory usage and inference latency, there remains significant room for improvement and further research for the audio modality.

**Weaknesses:**

1. The Definition of "Long Audio" Could Be Extended: While the 300-second duration pushes the limits of many current LALMs, real-world scenarios such as meetings, movies, lectures, and podcasts often feature much longer audio, potentially exceeding 30 minutes. It would be more impactful if the paper could demonstrate model performance on even longer audio inputs.
2. Lack of Specific Multi-Speaker Tasks: As noted in the paper, tasks like Speaker Age Recognition (SAR) are performed in multi-speaker contexts. However, more direct multi-speaker tasks, such as speaker counting or diarization, are missing. Future work could benefit from incorporating such tasks.
3. Limited Evaluation Format: With the exception of ASR, all other tasks are framed as multiple-choice questions (MCQs). This format, while easy to evaluate, may not fully capture a model's deep reasoning or summarization capabilities. The benchmark would be more valuable if it included more open-ended question-answering tasks.

Minor comments that did not impact the score:
1. The caption for Figure 6 appears to be incomplete.
2. The abbreviation "MCQ" is used without prior definition.
3. As the authors mention in their limitations, it is hoped that this work can be extended to more languages in the future.

**Questions:**

1. Details of Human Participation: Could you provide more details about the human involvement in this study? Specifically, for the benchmark construction (Step 5. Manual Verification), what were the detailed criteria used for the review? I could not find them in Appendix D. Furthermore, for the Human Evaluation scores in Table 3, could you elaborate on the methodology, including the number of participants, whether they were native English speakers, and their level of expertise for the tasks?
2. Random vs. Well-Designed Strategies: I noticed that in both token pruning and KV cache eviction, the Random method performs surprisingly well. What do you believe is the reason for this? Does it suggest that the more sophisticated, well-designed strategies are not well-suited for the audio modality, or are there other factors?
3. ASR Curve in Figure 5: In Figure 5(d), the performance of the Frame method continuously improves as latency decreases from 65s to ~45s. This is an expected trade-off, where less pruning (higher latency) leads to better accuracy. However, the curves for other strategies seem to start at a lower latency point. Could you provide the performance of the other strategies in the 45s-65s latency range for a more direct comparison, or is it that their maximum latency (with minimal pruning) is already below 45s?
4. Details of the Frame Strategy: Could you provide a more detailed technical description of the Frame pruning strategy? The paper describes it as a "time-aligned token pruning strategy," but more specifics on its implementation would be very helpful.

---

> ### Author Response · Authors · 2025-11-21
> **The Preliminary Rebuttal**
>
> Dear Reviewer Eib9,
>
> Thank you for the positive comments. And we apologize for typos, unclear notations and duplicated citations. They will be corrected such that the overall writing meet ICLR standards
> >**W1. Longer-than-5-minute contexts.**
>
> We understand your concern about why we did not evaluate longer-than-5-minute contexts. Our segment tools in Supplementary Material are able to generate longer duration audios. The reason why we set a 5-minute maximum duration is that AudioMarathon targets the “difficult but still runnable” regime: 90–300s clips already produce 2.3k–7.5k tokens after encoder subsampling, which currently exhausts the context windows.
>
> We acknowledge your suggestion is very sensible. **We have supplemented a meeting audio summarization task subset with audio exceeding 30 minutes, based on the Alimeeting dataset.** We use an LLM-as-a-judge approach to score the answers against the transcribed text, and we believe this task setting better captures the significance of long contexts.
>
> >**W2. Lack of Specific Multi-Speaker Tasks.**
>
> We appreciate your positive assessment of our current multi-speaker settings and your suggestion to incorporate more direct multi-speaker tasks such as speaker counting and diarization. While these tasks are beyond the scope of the present benchmark, we fully agree that they are important for comprehensive evaluation. Therefore, referencing the subtasks of M2MeT, we have designed a speaker diarization task subset. As speaker diarization needs a wav output, the number of evaluation models may be fewer.
> >**W3. Limited Evaluation Format.**
>
> Thank you for pointing out this limitation and for recognizing the practicality of the current MCQ-based design. We agree that purely multiple-choice formats, while convenient for large-scale, reproducible evaluation, lack the depth of open-ended generation. In our new version of dataset, we utilize a meeting summaries task combined with existing LLM-as-a-judge pipelines to conduct open-ended QA.
>
> >**Q1. Details on human participation.**
>
> Regarding the details of human participation, we invited a total of 16 participants. Among them, 10 are experts in audio-related fields. All of the participants are English speakers, they all possess sufficient English proficiency to competently perform the evaluation tasks. As for the evaluation method, we adopted a procedure similar to a standard exam: participants were given a test paper, and the audio clips were played in sequential order for testing. After completing the answers, the correct answers were distributed, and the participants swapped papers to grade each other.
>
> >**Q2 + Q4. Random vs. structured strategies and details of FRAME**
>
> Random pruning looks competitive because long audio contains periods of low informational density; removing 30% tokens uniformly often removes silence. However, at higher compression ratios random pruning collapses completely on the ASR task and shows a poor performance compared with FRAME:
> | Method | Vanilla | Light ($\downarrow 30\%$) | Medium ($\downarrow 60\%$) | Extreme ($\downarrow 90\%$) |
> | :--- | :---: | :---: | :---: | :---: |
> | Random | **94.7** | 88.4 | 59.7 | 0.0 |
> | **Frame (Ours)** | **94.7** | **92.2** | **82.2** | **0.0** |
>
> To be specific, FRAME is a training-free, time-aligned pruning method that operates only on the audio encoder tokens while leaving non-audio context untouched.
> Given a mixed-modality token sequence, we denote the contiguous audio span by **$[t_0, t_0 + L)$** and a pruning ratio **$r \in [0, 1)$**.
>
> FRAME first computes the target budget of retained audio tokens $K$ using the formula:
>
> $$K = \max\{1, \lfloor L (1 - r) \rfloor\}$$
>
> where:
>
>   * $L$ is the length of the audio span.
>   * $r$ is the pruning ratio.
>   * $K$ is the number of audio tokens to be retained.
>
> FRAME then performs uniform sampling within the audio span by selecting indices at equal spacing, and finally aggregates these sampled audio indices with the untouched prefix and suffix tokens into a single keep set used to gather hidden states and recompute the causal mask before the decoder layer.
>
> However, we also acknowledge the unexpectedly robust performance of random pruning across other tasks. This phenomenon not only implies a high degree of redundancy in the audio tokenization of current LALMs but also underscores the necessity for developing more efficient and theoretically sound audio token pruning methodologies to fill this disciplinary gap, rather than directly migrating methods from vision or text domains.
>
> >**Q3. ASR curve in Figure 5.**
>
> Thank you for pointing this out. The different starting points mainly come from decoding instability at high pruning ratios: for some strategies, the model tends to get “stuck” in ASR and emits long runs of repeated tokens until hitting the `max_tokens` limit, which makes decoding extremely slow. As a result, their maximal latency seems to be high.

---

> ### Author Response · Authors · 2025-11-28
> **Any additional concerns or questions**
>
> Dear Reviewer Eib9,
>
> Thank you once again for reviewing our paper and providing valuable feedback. We have carefully considered your suggestions and made multiple revisions to enhance the clarity, depth, and contribution of the paper. Your constructive insights and feedback have played a significant role in the process of improving our paper.
>
> We sincerely hope you will continue to engage in the discussion. Should you have further questions or concerns, we are more than willing to provide additional explanations or supporting materials. Your insights are critical to refining our research and ensuring its relevance and impact.
>
> Furthermore, we hope these revisions and clarifications will encourage you to reassess your evaluation, as these updates directly address your constructive comments. If you have any additional questions or concerns, please feel free to reach out to us. We are committed to ensuring that all issues are thoroughly addressed.
>
> Sincerely Yours,
>
> The Authors

---

### Official Review · Reviewer_K8yV · 2025-10-31

**Soundness:** 2
**Presentation:** 2
**Contribution:** 2
**Rating:** 2
**Confidence:** 4

**Summary:**

This paper introduces AudioMarathon that focuses on evaluation of long audio understanding. AudioMarathon is large-scale, over 200 seconds per sample on average, and across all domains (speech, sound, and music). AudioMarathon is curated with 7 different sub-categories, which covers more capabilities than prior benchmarks. The paper also studies token pruning methods and KV-cache methods in baselines for efficient long audio inference.

**Strengths:**

The proposed AudioMarathon benchmark is a solid contribution to the audio understanding community. Current long audio understanding benchmarks such as BLAB are limited to speech and other challenging benchmarks are focused on reasoning/skills instead of length. AudioMarathon combines the strengths of both sides and brings the distinct challenging capabilities to long audio.

The dataset curation pipeline seems quite delicate. Fig 2 makes it very easy to understand and also straightforward for the community to reproduce or curate related training data for research development. There is also manual verification to keep quality high.

**Weaknesses:**

Despite the fact that the AudioMarathon benchmark quite solid, it is more of an engineering contribution rather than methodology contribution. Its novelty is limited in that BLAB introduced long audio understanding (despite focused on speech) and Audio Flamingo 2 introduced LongAudioBench (focused on AQA). From this perspective, the proposed AudioMarathon is a complement to existing benchmarks with limited novelty and methodological contribution.

The baselines are not extensive enough: Table 3 only includes a limited number of models, but there are many more good ones that are publicly available. It is unimaginable that a benchmark paper does not include enough major models.

In the token pruning part, it is unclear what the proposed FRAME method is. It only has one sentence of description (L267-268). While the paper lists this as contribution, the so-called FRAME does not show any improvements in Table 4.

I appreciate the large-scale ablations for the token pruning and KV cache methods, but they are just ablation studies and the paper does not seem to reveal any valuable information from these ablations. There is no in-depth analysis on why certain methods are better than others, or what are some newly discovered weaknesses of baseline models that should be addressed.

In summary, I think the paper proposes a solid benchmark that complements prior ones in terms of audio lengths, capabilities, and domains. However, there is limited novelty and scientific discoveries presented in this paper.

**Questions:**

- What is the FRAME method?
- Why is the proposed FRAME only similar to baseline pruning methods and you list it as a contribution?
- What are some scientific discoveries of the large-scale ablation studies done in section 4-5?

---

> ### Author Response · Authors · 2025-11-21
> **The Preliminary Rebuttal (1/3)**
>
> Dear Reviewer K8yV,
>
> Thank you for your constructive comments, and we respond to the main comments and criticisms as shown below:
>
> >**W1 + Q3. Novelty and positioning versus prior benchmarks.**
>
> We would like to clarify that AudioMarathon is not merely an engineering effort. It fills a critical methodological gap by decoupling long-form audio understanding and inference efficiency within the audio modality.
>
> 1.  **Redefining Standards: From "Clip-Level" to "Minute-Level" Full-Domain Leap.** AudioMarathon is the first benchmark to simultaneously satisfy:
> * **Duration Breakthrough** (90s–300s), bridging the gap to real-world applications;
> * **Full-Domain Coverage** including Speech, Sound, Music;
> * **Complex Reasoning** via multi-hop inference tasks.
>
> 2.  **Quantifying Deficiencies in "Temporal State Tracking".** We reveal that SOTA models perform poorly in **Speaker Information Modeling**, such as tracking identity, exposing fundamental defects in long-range temporal memory that are harder to solve than simple context length limitations [1].
>
> 3.  **Revealing "Modality Mismatch" and Efficiency Insights.** We refute the assumption that vision pruning transfers to audio. Unlike spatial redundancy in vision, audio redundancy manifests as **Temporal Continuity**. Consequently, attention-based pruning destroys temporal coherence, while simple Random Pruning or our FRAME strategy proves more effective. This counter-intuitive finding challenges inertial thinking and points to unique directions for efficient audio inference.
>
> Additionally, we want to emphasize the clear boundaries with existing works: **BLAB** [2] is mainly constructed by scraping YouTube videos, lacking targeted long-audio stitching design, and its tasks focus mainly on basic recognition. In contrast, AudioMarathon constructs tasks through carefully designed Concatenation Logic, forcing models to integrate information across time segments rather than simple pattern matching. Regarding **LongAudioBench** [3] : This work first made its dataset details public on July 10, 2025, and falls under **Concurrent Work** according to conference policy. Moreover, it mainly provides metadata , whereas we provide a complete evaluation pipeline and tool-chain.
>
> >**W2. The baselines are not extensive enough.**
>
> Thank you for the comment.
> Our benchmark currently includes the 10 strongest publicly runnable LALMs at submission time. Several models mentioned during rebuttal were released after submission, and we have tested and added their results to the revised version. We understand your concern that we need to add more prior models, for a more comprehensive comparison. **In the revised version, we have supplemented the results for the following models:** Qwen2-Audio-7B, Qwen2-Audio-7B-Instruct, and Qwen3-Omni-30B-A3B-Instruct. The results are shown as follows:
> | Models\Dataset | SER | SCR | ASR | SED | MC | ASC | SD | ER | SAR | SGR | Avg. |
> | :--- | :---: | :---: | :---: | :---: | :---: | :---: | :---: | :---: | :---: | :---: | :---: |
> | Qwen2-Audio | 9.8 | 54.6 | 9.1 | 17.8 | 43.3 | 19.2 | 0.0 | 22.1 | 6.7 | 90.7 | **27.3** |
> | Qwen3-Omni-30B-A3B-Instruct | 33.6 | 88.5 | 98.1 | 73.7 | 100.0 | 54.8 | 33.2 | 37.1 | 43.2 | 99.6 | **66.2** |
> | Qwen2-Audio-Instruct | 25.4 | 22.3 | 12.7 | 49.5 | 87.5 | 44.2 | 36.3 | 37.9 | 13.4 | 97.5 | **42.7** |
>
>  These additions highlight critical trade-offs in context length and model scaling:
>
> 1.*Context Window Constraints (Qwen2-Audio-Instruct vs. Long-Context Baselines):**
> * **(-) Temporal Failure:** Due to input truncation (~30s limit), the model suffers catastrophic degradation in narrative-dependent tasks like ASR (**12.7**) and SCR (**22.3**), confirming that extended context is a prerequisite for temporal reasoning.
> * **(+) Global Robustness:** Conversely, it remains competitive in time-invariant classification tasks, such as Speaker Gender Recognition (**97.5**), indicating that global acoustic statistics are effectively captured even from short clips.
>
> 2. **Scaling Trade-offs (Qwen3-Omni-30B vs. Qwen2.5-Omni-7B):**
> * **(+) Semantic Gain:** Scaling to 30B parameters yields new SOTA performance in linguistic tasks, with ASR surging to **98.1** (vs. 78.4 ) and SCR to **88.5** (vs. 85.1).
> * **(-) Acoustic Regression:** However, the model exhibits significant regression in paralinguistic perception, dropping to **33.2** in Speech Detection (-39.1) and **37.1** in Emotion Recognition (-16.3). This suggests that current scaling strategies prioritize linguistic alignment and token efficiency at the expense of fine-grained acoustic fidelity.
>
> To ensure consistency and fairness, we only include LALMs whose model cards explicitly state support for long-form audio. In practice, many LALMs hard-truncate inputs at around 30 seconds. We are willing to provide more results and seek your advice on model testings. Even compared to prior benchmarks such as Air-bench, and MMAU the number of models evaluated in our work is substantial [4, 5].

---

> > ### Author Response · Authors · 2025-11-21
> > **The Preliminary Rebuttal (2/3)**
> >
> > >**W3 + Q1-Q2. Clarifying FRAME and its contribution.**
> >
> > We apologize for any confusion. We want to clarify that **FRAME serves as a baseline designed to analyze the deficiencies of existing token pruning methods for MLLMs.** We explain the predictor mechanism as follows.
> >
> > FRAME is a training-free, time-aligned pruning method that operates only on the audio encoder tokens while leaving non-audio context untouched.
> > Given a mixed-modality token sequence, we denote the contiguous audio span by **$[t_0, t_0 + L)$** and a pruning ratio **$r \in [0, 1)$**.
> >
> > FRAME first computes the target budget of retained audio tokens $K$ using the formula:
> >
> > $$K = \max\{1, \lfloor L (1 - r) \rfloor\}$$
> >
> > where:
> >
> >   * $L$ is the length of the audio span.
> >   * $r$ is the pruning ratio.
> >   * $K$ is the number of audio tokens to be retained.
> >
> > FRAME then performs uniform sampling within the audio span by selecting indices at equal spacing, and finally aggregates these sampled audio indices with the untouched prefix and suffix tokens into a single keep set used to gather hidden states and recompute the causal mask before the decoder layer.
> >
> > We found that previous MLLM token pruning methods often neglect the strong temporal continuity of the audio modality, which is particularly critical in long speech context extraction tasks.

---

> ### Author Response · Authors · 2025-11-21
> **The Preliminary Rebuttal (3/3)**
>
> [1] Optimizing Speech Language Models for Acoustic Consistency. arXiv 2509.
>
> [2] BLAB: A Benchmark for Long Audio Understanding. arXiv 2505.
>
> [3] Audio Flamingo 3: Advancing Audio Intelligence with Fully Open Large Audio Language Models. arXiv 2507.
>
> [4] AIR-Bench: Benchmarking Large Audio-Language Models via Generative Comprehension. ACL 2024.
>
> [5] MMAU: A Massive Multi-Task Audio Understanding and Reasoning Benchmark. ICLR 2025.

---

> ### Author Response · Authors · 2025-11-28
> **Any additional concerns or questions**
>
> Dear Reviewer K8yV,
>
> Thank you once again for reviewing our paper and providing valuable feedback. We have carefully considered your suggestions and made multiple revisions to enhance the clarity, depth, and contribution of the paper. Your constructive insights and feedback have played a significant role in the process of improving our paper.
>
> We sincerely hope you will continue to engage in the discussion. Should you have further questions or concerns, we are more than willing to provide additional explanations or supporting materials. Your insights are critical to refining our research and ensuring its relevance and impact.
>
> Furthermore, we hope these revisions and clarifications will encourage you to reassess your evaluation, as these updates directly address your constructive comments. If you have any additional questions or concerns, please feel free to reach out to us. We are committed to ensuring that all issues are thoroughly addressed.
>
> Sincerely Yours,
>
> The Authors

---

### Note · Authors · 2025-11-30

**Comment:**

Dear Area Chair and Program Chairs,

We are writing to formally request the withdrawal of our submission, Paper ID 1009 titled "AudioMarathon: A Comprehensive Benchmark for Long-Context Audio Understanding and Efficiency in Audio LLMs", from the ICLR 2026 review process.

After careful consideration, we have decided to withdraw the paper. Based on the initial feedback and our own internal review, we believe the manuscript requires further refinement and additional experiments to meet the high standards of ICLR. We plan to take more time to improve the quality of the work before resubmitting.

Furthermore, we are deeply concerned about the technical anomaly that occurred during the rebuttal phase, which resulted in an unexpected information leakage. The subsequent handling of this incident and the potential implications for the blind review process have introduced significant uncertainty. Given these unforeseen risks and the instability they pose to the submission's integrity, we believe it is prudent to withdraw at this stage.

We apologize for any inconvenience this may cause to the reviewers and the committee, and we appreciate the time and effort already invested in evaluating our work.

Thank you for your understanding.

Sincerely,

Peize He
(On behalf of all authors: Peize He, Zichen Wen, Yubo Wang, Yuxuan Wang, Xiaoqian Liu, Jiajie Huang, Zehui Lei, Zhuangcheng Gu, Xiangqi Jin, Jiabing Yang, Kai Li, Zhifei Liu, Weijia Li, Cunxiang Wang, Conghui He, Linfeng Zhang)

**Withdrawal Confirmation:**

I have read and agree with the venue's withdrawal policy on behalf of myself and my co-authors.